# A population of Vasa2 and Piwi1 expressing cells generates germ cells and neurons in a sea anemone

Paula Miramón-Puértolas [ORCID], Eudald Pascual-Carreras [ORCID] & Patrick R. H. Steinmetz [ORCID] [✉]

Germline segregation, essential for protecting germ cells against mutations, occurs during early embryogenesis in vertebrates, insects and nematodes. Highly regenerative animals (e.g., cnidarians), however, retain stem cells with both germinal and somatic potentials throughout adulthood, but their biology and evolution remain poorly understood. Among cnidarians (e.g., sea anemones, jellyfish), stem cells are only known in few hydrozoans (e.g., *Hydra*). Here, we identify and characterize a rare, multipotent population of stem and/or progenitor cells expressing the conserved germline and multipotency proteins Vasa2 and Piwi1 in the sea anemone *Nematostella vectensis*. Using *piwi1* and *vasa2* transgenic reporter lines, we reveal that the Vasa2+/Piwi1+ cell population generates not only gametes, but also a diversity of proliferative somatic cells, including neural progenitors, in juveniles and adults. Our work has uncovered a multipotent population of Vasa2+/Piwi1+ stem/progenitor cells that forms the cellular basis to understand body plasticity and regenerative capacities in sea anemones and corals.

In vertebrates, insects and nematodes, primordial germ cells (PGCs) segregate from somatic lineages by maternal or zygotic induction during early embryogenesis[1]. PGCs then migrate into the developing gonad and establish a population of unipotent germline stem cells (GSCs)[2]. Early germline segregation is thus found predominantly in animals with low regenerative potential. In contrast, highly regenerative animals such as planarian flatworms, acoels, or hydrozoan cnidarians (Fig. 1A) exhibit a pool of pluri- or multipotent adult stem cells (ASCs), termed 'primordial stem cells' (PriSCs) hereafter, that retain both somatic and germinal potential throughout their lifetime[3–8]. As PriSCs are molecularly underexplored and found so far only in animals across a restricted set of scattered animal taxa, their evolutionary origin remains unclear[9–11].

Among cnidarians, the sister group of bilaterians (Fig. 1A), knowledge on germline specification and stem cell biology is restricted to hydrozoans, especially to the well-studied hydrozoans *Hydra sp.* and *Hydractinia sp. Hydra* possesses two unipotent epithelial stem cell populations and a pool of multipotent PriSCs, the interstitial stem cells (i.e., i-cells), which generate germ cells and somatic cell types including neurons, gland cells and stinging cells (cnidocytes)[4,12]. In *Hydractinia*, pluripotent i-cells generate all lineages, including epithelial cells[13]. As stem cells have remained elusive in any other cnidarian taxa, i-cells have been considered a hydrozoan-specific trait.

Anthozoans (i.e., corals and sea anemones; Fig. 1A) form the sister group to all other cnidarians and reproduce both sexually and asexually. So far, approaches using transgenic lines or single-cell transcriptome sequencing of post-larval stages have failed to clearly identify any anthozoan PriSCs[14–19]. However, their existence is conceivable given the high regeneration potential of anthozoans during asexual reproduction or after bisection[20–22]. Furthermore, recent reports showed that gametes and somatic tissue share spatially restricted, post-larval genomic mutations in scleractinian corals[23,24]. We therefore aimed to identify potential PriSCs in the sea anemone *Nematostella vectensis* by using a combination of spatial expression analysis and immunostainings of conserved germline and multipotency marker genes (e.g. *piwi, vasa*), and newly generated transgenic reporter lines.

Michael Sars Centre, University of Bergen, Thormøhlensgt. 55, N-5008 Bergen, Norway. [✉]e-mail: patrick.steinmetz@uib.no

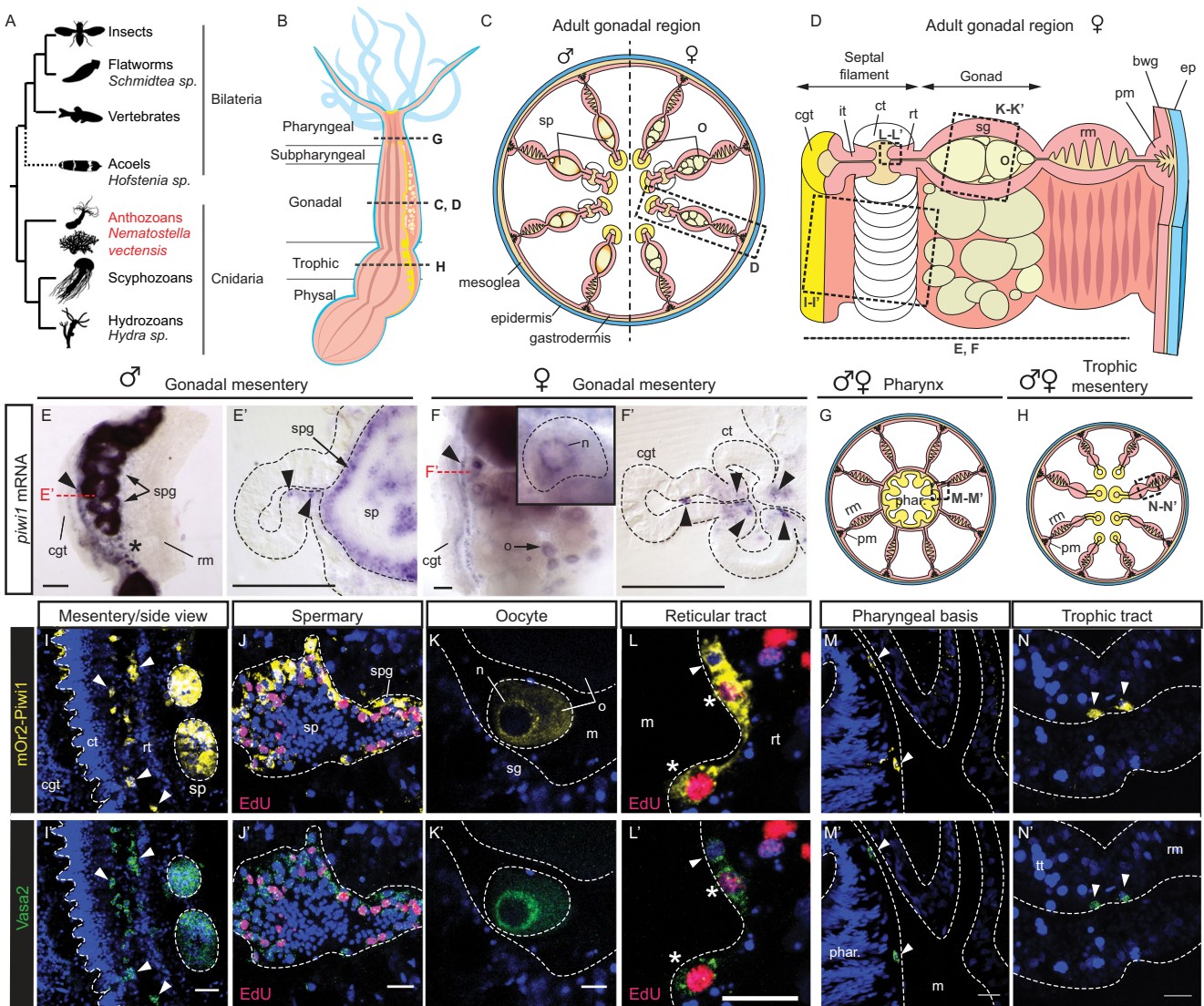

**Fig. 1 | Vasa2 and mOr2-Piwi1 proteins colocalize to developing gametes and extra-gonadal cells in the mesentery. A** Simplified phylogenetic tree highlighting the position of the sea anemone *Nematostella vectensis* and other relevant animal taxa. All animal silhouettes are licensed under CC0 1.0 Universal Public domain and taken from https://www.phylopic.org.**B–D, G, H** Schematics of longitudinal section of the adult polyp (**B**) and of cross sections (**C, D, G, H**) of pharynx (**G**), trophic region (**H**), body column (**C**) or single mesentery (**D**) within the gonadal region (see B). Orientation: Oral towards top. **E-F'** *piwi1* whole mount ISH in dissected gonadal region tissue pieces (**E, F**) and their cross sections (**E', F'**). *piwi1* expression found in spermatogonia (**E, E'**; spg), perinuclear granules ('nuage') of oocytes (**F**, inlet; 'n') and in basiepithelial cells concentrating at the basis of the septal filament (**F**, arrowheads). (**I-N'**) Confocal imaging stacks of gonadal mesentery region tissues immunostained for mOr2-Piwi1 (**I-N**; yellow), Vasa2 (**I'-N'**, green), and labelled for S-phase by EdU (**J, J', L, L'**; 3 days pulse, red) viewed from lateral (**I, I'**) or cross sectioned (**J-N'**) as indicated in (**B–D, G, H**). Vasa2 and mOr2-Piwi1 colocalize to spermatogonia cells (**I-J'**; spg) and the nuage of oocytes (**K, K'**; 'n'). Outside of the gonad, cells colocalizing Vasa2 and mOr2-Piwi1 (white arrowheads) were found along the septal filament basis (**L, L'**), the pharyngeal basis (**M, M'**) and the trophic tract (**N, N'**). Note that EdU label marks nuclei in Vasa2+/Piwi1+ peripheric spermatogonia (**J, J'**; spg) but is scarce in more centrally located, differentiated, Vasa2−/Piwi1− spermatocytes (sp). Some basiepithelial cells were also EdU+ (**L, L'**; asterisks). Blue: Hoechst DNA dye. bwg body wall gastrodermis, cgt cnidoglandular tract, ct ciliated tract, ep epidermis, gam gametes, it intermediate tract, m mesoglea, o oocyte, pm parietal muscle, rm retractor muscle, rt reticular tract, sg somatic gonad, sp sperm, spg spermatogonia. Scale bars: 100 μm (**E-F'**), 20 μm (**I**) and 10 μm (**J'−N'**). Experiments were independently performed four times (**E-F'**) or twice (**I-N'**) with similar results.

In this work, we show that Vasa2+/Piwi1+ cells constitute a multipotent population of stem and/or progenitor cells from which gametes and a diversity of somatic cells, including neurons, develop in juvenile and adult *Nematostella*.

## Results

### Germline multipotency program genes expressed beyond the gonad

PriSCs, PGCs and GSCs share the expression of an evolutionarily conserved gene set, the germline multipotency program (GMP)[25]. Key GMP components comprise RNA helicases (Vasa and Pl10), RNA-binding (Tudor, Nanos, Pumilio, Boule and Bruno) and RNA-cleaving proteins (Piwi). These form ribonucleoprotein aggregates with target mRNAs discernable as electron-dense fibrillar structures (i.e. germline granules or 'nuage') on an ultrastructural level[25–27]. Their consistent expression in PriSCs of diverse animal species (e.g., in planarian and acoel neoblasts, hydroid cnidarian i-cells, sponge archeocytes and choanocytes) makes them interesting as marker genes to identify germline and PriSCs in uncharacterized animals[7,10,25,28–31]. We therefore used in situ hybridization of the *Nematostella* GMP gene orthologs *piwi1, piwi2, vasa1, vasa2, pl10* and *tudor* to study the expression of these genes in adult male and female mesenteries, which are inner

epithelial folds that harbour the developing gametes (Fig. 1B–F' and Supplementary Fig. 1). We confirmed that previously reported *piwi1, piwi2, vasa1* and *vasa2*[32] and additionally *tudor* and *pl10* genes are expressed in developing oocytes and in spermatogonia at the periphery of the spermaries (arrows in Fig. 1E–F' and Supplementary Fig. 1; Source Data). In addition, we found *piwi1, piwi2, pl10* and *tudor* expression in basiepithelial cells of the septal filament, the tip of the mesentery located distal to the gonadal tract (Fig. 1D–F' and Supplementary Fig. 1; arrowheads; Source Data). Based on their location, these cells are reminiscent of the putative PGCs previously identified in *Nematostella* by Vasa2 immunostainings[33].

**Piwi1 and Vasa2 protein colocalize to gametes and somatic cells**
We confirmed that previously published monoclonal[32] and polyclonal *Nv*-Vasa2 antibodies[33] consistently label developing oocytes and spermatogonia (Supplementary Fig. 2). Labelling of Vasa2+ cell duplets by the thymidine analogue 5-ethynyl-2'-deoxyuridine (EdU), which integrates into DNA during S-phase, indicates the presence of potential oogonia within the gonad epithelium (Supplementary Fig. 3A, B, D). In addition to gametes, both antibodies co-labelled basiepithelial Vasa2+ cells (Supplementary Fig. 2; arrowheads), partly labelled by EdU (Supplementary Fig. 3B-C; asterisks), which notably were not restricted to the gonadal region of the mesentery (Fig. 1B–D). Instead, they were distributed along nearly the entire oral-aboral extent of the mesentery (Supplementary Fig. 3E-I''). In both males and females, these cells concentrated near the mesoglea along the reticular tract (Fig. 1D; 'rt'), a poorly described part of the septal filament, and near the distal end of the retractor muscle (Supplementary Fig. 2 and 3B, C; arrowheads). In other sea anemones, the reticular tract consists of endocytic cells characterised by vacuoles and/or spherical concretions (hence 'reticular')[34].

To test if Vasa2+ cells co-express Piwi1 protein, as suggested by *piwi1* mRNA expression (Fig. 1E, F'), we generated a CRISPR/Cas9-mediated transgenic knock-in allele (*piwi1mOr2*) resulting in an mOrange2 fluorophore fused to the N-terminus of the Piwi1 protein (mOr2-Piwi1; Supplementary Fig. 4A). Co-immunodetection confirmed the colocalization of mOr2-Piwi1 and Vasa2 proteins to EdU+, putative spermatogonia and growing oocytes (Fig. 1I-K') of adult polyps. Within developing oocytes, mOr2-Piwi1 and Vasa2 proteins colocalize with *tudor* and *piwi1* mRNAs to perinuclear germ granules ('n'; Fig. 1F (inlet), K, K' and Supplementary Fig. 1E(inlet), 3J-K)[32,33]. In addition, we observed that high levels of mOr2-Piwi1 protein ([mOr2-Piwi1]$_{high}$ hereafter) colocalize with Vasa2 protein to basiepithelial cells along the reticular tract all along the mesentery (Fig. 1I, I', L-N'). In juveniles, Vasa2 and [mOr2-Piwi1]$_{high}$ co-localized to similar basiepithelial cells that were restricted to a narrow region along the mesenteries from the pharyngeal basis into the trophic region (Fig. 2A–C', E-F', H-I'). Consistent with Vasa2 and mOr2-Piwi1 protein co-localization, *piwi1* and *vasa2* mRNA transcripts were also restricted to basiepithelial cells within the same regions (Fig. 2D, G, J; arrowheads; Source Data).

Intrigued by the scarcity of Vasa2+/[mOr2-Piwi1]$_{high}$ cells detected by confocal imaging, we quantified their proportion in fed juvenile and adult *piwi1mOr2/+* polyps using flow cytometry (Fig. 2K; see Supplementary Fig. 5 and Methods for gating strategies). We could separate two populations of mOr2-Piwi1+ cells with clearly distinct signal intensities (Supplementary Fig. 5; 'Low' and 'High'). We assume that the small subset of mOr2-Piwi1+ cells with ~10-fold higher signal intensities compared to the bulk of mOr2-Piwi1+ cells largely correspond to [mOr2-Piwi1]$_{high}$ cells identified by confocal microscopy. In fed juveniles, [mOr2-Piwi1]$_{high}$ cells represent only $0.04 \pm 0.01\%$ (i.e., ~4 in 10.000 cells) of all polyp cells, while their mean proportion corresponds to $0.39 \pm 0.25\%$ in adult females (Fig. 2K). This ~10-fold increase is at least partly explained by the presence of [mOr2-Piwi1]$_{high}$ small oocytes (<50 μm; see Methods) in adults but not juveniles.

As Vasa2+/[mOr2-Piwi1]$_{high}$ cells express GMP genes, are relatively small and are located basiepithelially, they present several hallmarks of hydrozoan interstitial stem cells (e.g., *Hydra* i-cells)[5,30,35,36]. We therefore tested their proliferative potential using EdU labelling. Using confocal imaging, we showed that a subset of adult basiepithelial, Vasa2+/[mOr2-Piwi1]$_{high}$ cells are EdU+ and thus proliferative (Fig. 1L-L'; asterisks). Notably, however, some cells remained unlabeled even after a 3-days long EdU pulse (Fig. 1L-L'; arrowhead). Quantification by flow cytometry (Supplementary Fig. 5 and Methods) confirmed that indeed only $41.5 \pm 11.5\%$ of all [mOr2-Piwi1]$_{high}$ cells in adults are EdU+ after a 3 days-long pulse (Fig. 2M). In contrast, the proportion of EdU+ cells among juvenile [mOr2-Piwi1]$_{high}$ cells is considerably higher ($72.9 \pm 3.65\%$), despite using a shorter, 1-day long EdU pulse (Fig. 2M). These results indicate that a higher proportion of adult Vasa2+/mOr2-Piwi1+ cells are slow cyclers or quiescent in comparison to juveniles. We conclude that Piwi1 and Vasa2 proteins are co-expressed in developing gametes and in a rare population of extra-gonadal, basiepithelial cells that are at least partly proliferative and form narrow lines along the entire mesenteries in juvenile and adult polyps.

Interestingly, we also observed cells in different parts of the juvenile body column containing mOr2-Piwi1 protein at levels near the detection limit ([mOr2-Piwi1]$_{low}$). These cells spread along the parietal muscle tract and in the body wall gastrodermis (Fig. 2I' and Supplementary Fig. 6B', C'; arrows), along the cnidoglandular tract basis (Supplementary Fig. 6D'; arrows) and throughout the epidermis (Supplementary Fig. 6A'; black arrowheads in Fig. 2I' and Supplementary Fig. 6C'). Interestingly, these cells were all devoid of detectable Vasa2 protein (Supplementary Fig. 6A–D) or *vasa2* mRNA (Fig. 2J). In principle, their [mOr2-Piwi1]$_{low}$ could have resulted from *piwi1-mOr2* mRNA expression outside of Vasa2+/[mOr2-Piwi1]$_{high}$ cells. We tested this hypothesis but found no trace of *piwi1* (Fig. 2D, G) or *mOr2-piwi1* (Supplementary Fig. 7A, B', E-F) in any of these cells. We therefore concluded that the most likely scenario to explain our observations is that [mOr2-Piwi1]$_{low}$ cells are the progeny of Vasa2+/[mOr2-Piwi1]$_{high}$ cells from which they receive low amounts of mOr2-Piwi1 protein by cytoplasmatic inheritance (summarized in Fig. 6A, Supplementary Fig. 6E).

As cells with [mOr2-Piwi1]$_{low}$ appeared much more numerous than those with [mOr2-Piwi1]$_{high}$ in confocal images, we assessed their proportion by flow cytometry (Fig. 2L, Supplementary Fig. 5). We found that [mOr2-Piwi1]$_{low}$ cells represent $53.1 \pm 2.11\%$ of all cells in juveniles, and only $20.9 \pm 13.6\%$ of all cells in adults (Fig. 2L). In juveniles, nearly two-thirds ($63.3 \pm 2.65\%$) of [mOr2-Piwi1]$_{low}$ cells were EdU + after a 1 day-long pulse (Fig. 2N and Supplementary Fig. 6A''-D''; asterisks). In adults, however, only $41.3 \pm 8.47\%$ of [mOr2-Piwi1]$_{low}$ cells were EdU+ despite a much longer 3-days long pulse (Fig. 2N). [mOr2-Piwi1]$_{low}$ cells are thus not only less abundant in adults but also show a much lower EdU index compared to juveniles.

**Somatic derivatives of Vasa2+/[mOr2-Piwi1]$_{high}$ cell population**
Aiming to study the potential cell lineages of the extra-gonadal Vasa2+/[mOr2-Piwi1]$_{high}$ cell population, we generated a CRISPR/Cas9-mediated *piwi1P2A-GFP* knock-in allele (Supplementary Fig. 4B). The P2A self-cleaving peptide[37,38] located between the GFP and the N-terminal Piwi1 region allows cytoplasmic inheritance of GFP in a Piwi1-independent manner. This *piwi1P2A-GFP* reporter line enabled us to track the transiently GFP+ progeny of Vasa2+/[mOr2-Piwi1]$_{high}$ cells. A *piwi1P2A-GFP*/*piwi1mOr2* double reporter line confirmed co-localization of [GFP]$_{high}$ and [mOr2-Piwi1]$_{high}$ protein to developing oocytes (Fig. 3A-A', C-C'), potential oogonia in the gonad epithelium (Fig. 3A-A', C-C'; pink arrowheads) and Vasa2+ cells in the reticular tract of adults (Fig. 3A, B'; white arrowheads) and juveniles (Fig. 4A–D'). The successful separation of GFP and Piwi1 proteins from the *piwi1P2A-GFP* allele is demonstrated by the detection of GFP, but not mOr2, in the nucleus of basiepithelial cells in the reticular tract and in oocytes of *piwi1P2A-GFP*/

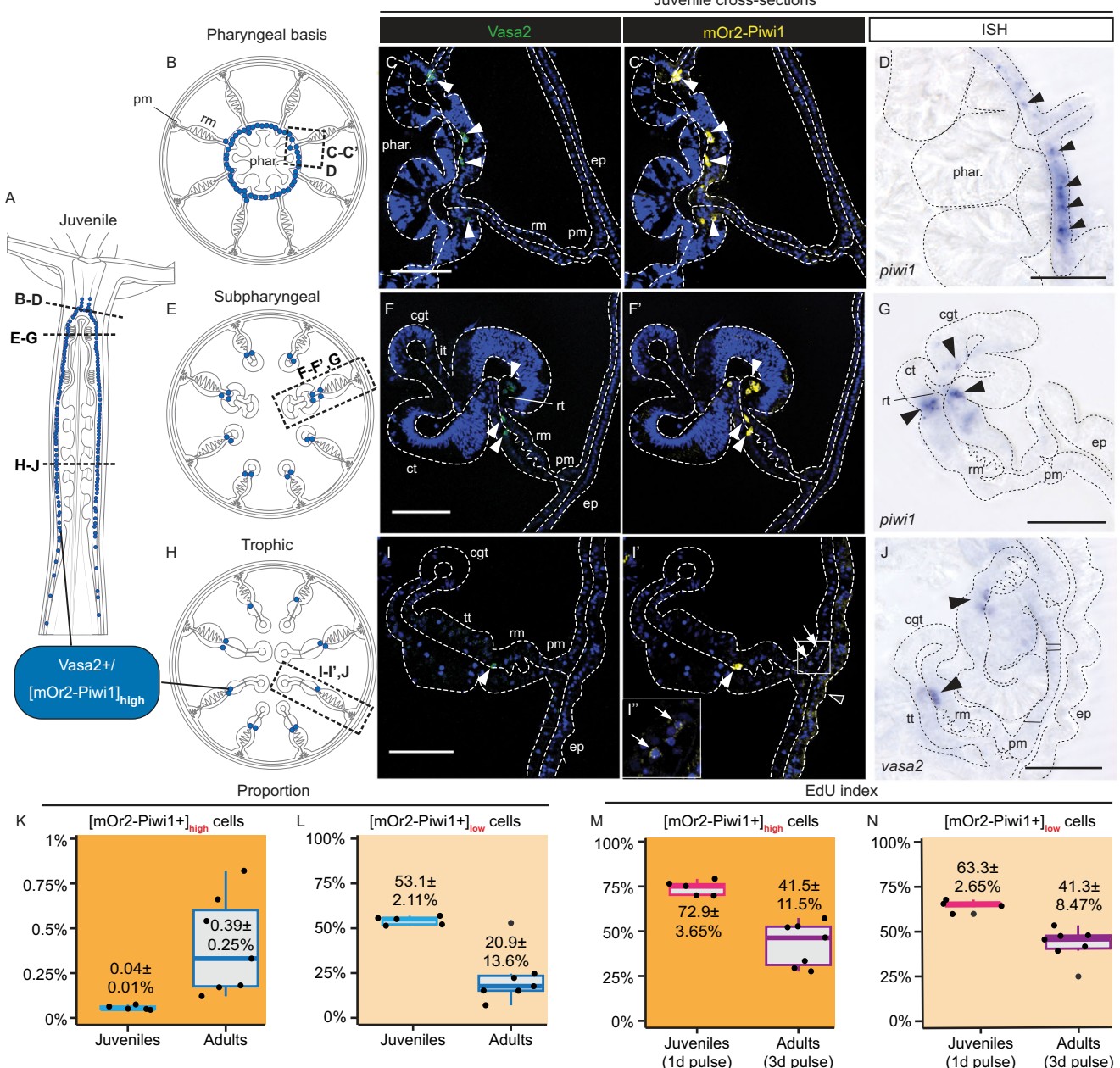

**Fig. 2 | Vasa2+/Piwi1+ basiepithelial cells locate along juvenile mesenteries, and represent a rare, proliferative cell population in juveniles and adults.**
**A**, **B**, **E**, **H** Schematics of longitudinal (**A**) and cross sections across the pharyngeal (**B**), subpharyngeal (**E**) and trophic (**H**) regions of a juvenile polyp. **C-C'**, **F-F'**, **I-I'** Confocal imaging stacks of cross-sectioned mesenterial and pharyngeal regions from *piwi1^mOr2* juvenile polyps immunostained for Vasa2 (**C**, **F**, **I**; green) and mOr2-Piwi (**C'**, **F'**, **I'**; yellow) as indicated in (**B**, **E**, **H**). Vasa2 and mOr2-Piwi1 colocalize to basiepithelial cells (arrowheads) at the basis of the pharyngeal gastrodermis (**C-C'**), the subpharyngeal region (**F**, **F'**) and the trophic regions (**I-I'**). mOr2-Piwi1 protein levels were close to the detection limit in basiepithelial cells of the parietal muscle and gastrodermal body wall regions of juveniles (**I'**; arrows), and in the epidermis (**I'**; black arrowhead); see also Supplementary Fig. 6. **D**, **G**, **J** Cross sections of whole-mount ISH of *piwi1* (**D**, **G**) and *vasa2* (**J**) genes reflect Vasa2 and mOr2-Piwi1 protein localization with *piwi1* expression in basiepithelial, gastrodermal cells at basis of the pharynx (**B**, **D**; arrowheads) and in the reticular tract of the subpharyngeal mesentery (**E**, **G**; arrowheads). *vasa2* expression in basiepithelial cells at the proximal and distal end of the trophic tract (**J**; arrowheads). **(K-N)** Proportion of [mOr2-Piwi1]_high or [mOr2-Piwi1]_low cells among all polyp cells (**K**, **L**), or of EdU+ cells among [mOr2-Piwi1]_high and [mOr2-Piwi1]_low cells (**M**, **N**) as determined by flow cytometry. Boxplots definition: see 'Data visualisation' and 'Methods'. Dots represent individual values. In (**K–N**): n(juveniles) = 5 distinct biological replicates consisting of a pool of 12 individuals. n(adults) = 7 distinct biological replicates consisting of individual females. Data are presented as mean values ± standard deviation. Source data are provided in Source Data file. Blue: Hoechst DNA dye. cgt cnidoglandular tract, ct ciliated tract, d day(s), ep. epidermis, it intermediate tract, phar pharynx, pm parietal muscle. rt reticular tract, rm retractor muscle, tt trophic tract. Scale bars: 50 μm (**C**, **F**, **I**). Experiments were performed independently twice (**C-C'**, **D**, **F-F'**, **G**, **I-I'**) or once (**J**) with similar results.

*piwi1^mOr2* polyps (compare Fig. 3B, C with Fig. 3B', C'). Interestingly, [GFP]_high/[mOr2-Piwi1]_low cells were observed adjacently to [GFP]_high/[mOr2-Piwi1]_high cells in the reticular tract (Fig. 4A, B'; double black arrowheads), suggesting equal transfer of cytoplasmatic GFP but not of the fusion protein mOr2-Piwi1 during cell division. Cells double-labelled by [GFP]_low and [mOr2-Piwi1]_low were also abundant in somatic regions of adults and juveniles. These cells, which potentially constitute progeny cells from Vasa2+/[mOr2-Piwi1]_high cells, locate to the

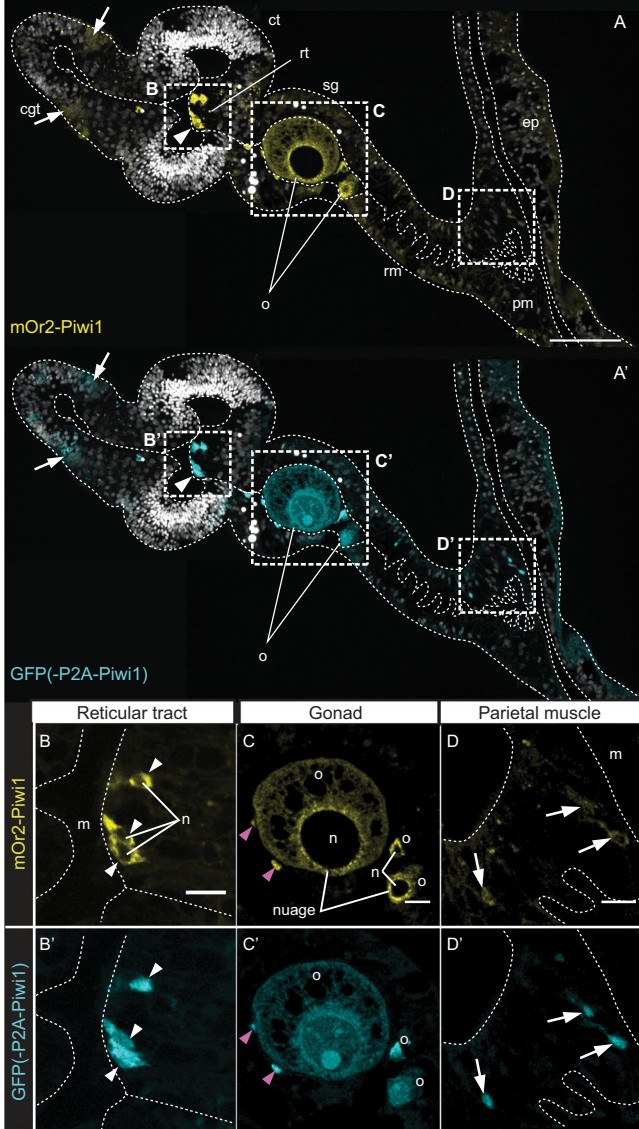

**Fig. 3 | Differential localisation detection levels of mOr2-Piwi1 and GFP in a** ***piwi1mOr2/P2A-GFP*** **CRISPR-KI double reporter line. A–D′** Confocal images of double transgenic *piwi1mOr2/P2A-GFP* polyps immunostained for mOr2 (**A–D**; yellow) and GFP (**A′–D′**; cyan). Cross sections of a female gonadal mesentery show high levels of mOr2 and GFP that colocalize to basiepithelial cells in the reticular tract (**A–C′**; arrowheads), and to oogonia (pink arrowheads) and developing oocytes ('o') in gonadal regions (**A-A′, C-C′**). Note cytoplasmic restriction of mOr2-Piwi1 in basiepithelial cells (**B**) and to perinuclear granules in oocytes (**C**; nuage). Extension of GFP localization to the cell nucleus (**B-C′**; 'n') validates the successful GFP-Piwi1 separation and the use of the *piwi1P2A-GFP* allele for lineage tracing. Low levels of GFP and mOr2 colocalize to cells at the basis of the cnidoglandular tract (**A-A′**; arrows at 'cgt'), retractor muscle (**A-A′**; arrow at 'rm'), and parietal muscle (**A-A′, D-D′**; arrows at 'pm') of the adult mesentery. Grey: Hoechst DNA dye. cgt cnidoglandular tract, ct ciliated tract, ep epidermis, it intermediate tract, m mesentery, n nucleus, o oocyte, pm parietal muscle, rm retractor muscle, rt reticular tract, sg somatic gonad. Scale bars: 50 μm (**A**) and 10 μm (**B–D**). All experiments were performed twice with similar results.

cnidoglandular tract base (Fig. 3A-A′; arrows in 'cgt') and along the parietal muscle region of adults (Fig. 3D-D′) and juveniles (Fig. 4C-D′). Few [GFP]_low/[mOr2-Piwi1]_low cells also intermingled between [GFP]_high/[mOr2-Piwi1]_high cells in the juvenile reticular tract (Fig. 4A; single arrowheads). This observation supports our assumption that [GFP]_low/[mOr2-Piwi1]_low cells get labelled by cytoplasmic inheritance

of Piwi1-mOr2 and GFP resulting in lower protein levels after division from a [GFP]_high/[mOr2-Piwi1]_high cell (Figs. 4E, 8A). Altogether, our analysis of juvenile and adult *piwi1P2A-GFP*/*piwi1mOr2* animals highlights an abundance of [GFP]_low/[mOr2-Piwi1]_low cells, likely derived from Vasa2+/[mOr2-Piwi1]_high cells, in different non-gonadal parts of the mesentery and body wall gastrodermis (Figs. 4E, 8A; Source Data). In line with the presence of epidermal [mOr2-Piwi1]_low cells (see above), we also found low, homogeneous levels of GFP throughout the epidermis of *piwi1P2A-GFP* animals (Supplementary Fig. 8A, B, E′, I), raising the possibility that epidermis derives from mesenterial Vasa2+/Piwi1+ cells. Alternatively, an independent, self-renewing population of epidermal cells with low, currently undetectable *piwi1* or *vasa2* gene expression levels could explain epidermal [mOr2-Piwi1]_low and [GFP]_low cells (see also discussion).

We further validated our assumption that proliferative Vasa2+/ [mOr2-Piwi1]_high cells give rise to a diversity of somatic progeny cells by generating an alternative *vasa2*::mOr2 reporter line using I-Sce I meganuclease-mediated transgenesis[39,40] (Supplementary Fig. 4C). We confirmed that the *vasa2*::mOr2 line reliably reports endogenous *vasa2* expression by successfully co-detecting [*vasa2*::mOr2]_high and Vasa2 protein in growing oocytes (Fig. 5A-A′), spermatogonia (Fig. 5B–B′) and along the narrow reticular tract, as previously described for juvenile and adult mesenteries (arrowheads in Fig. 5D–D′ and Supplementary Fig. 9A–E′, H-H′, K, L; Source Data). In addition, lower levels of *vasa2*::mOr2 were found in somatic, Vasa2– cells spread throughout the body column gastrodermis of juveniles (Supplementary Fig. 9A–G′, I–L; arrows & asterisks) and adults (Fig. 5C–F′). Their location, diversity, relatively low mOr2 protein levels and absence of detectable Vasa2 protein suggests that [*vasa2*::mOr2]_low cells are the progeny of Vasa2+/[mOr2-Piwi1]_high cells in adults (Figs. 5C and 8; Source Data) and juveniles (Supplementary Fig. 9K-L; Source Data). Many Vasa2–/[*vasa2*::mOr2]_low cells integrated EdU and thus continued to proliferate (Fig. 5D″–F″ and Supplementary Fig. 9F″–G″, I″–J″; asterisks). More specifically, Vasa2–/[*vasa2*::mOr2]_low cells were found in the cnidoglandular tract and the distal, highly proliferative part of the ciliated tract of juveniles (Supplementary Fig. 9F–G″)[41]. They also located along the retractor muscle, parietal muscle and the adjacent body wall gastrodermis in juveniles (Supplementary Fig. 9I-J′) and adults (Fig. 5E, F″). Notably, these gastrodermal cells often exhibited slim, neuron-like shapes (Fig. 5E′; arrow) or filopodia-like extensions as typical for migratory cells (Fig. 5F′; arrows).

We tested if putative progeny cells independently labelled by *piwi1P2A-GFP* or *vasa2*::mOr2 overlap by crossing both lines and analyzing their juvenile offspring. As expected, we found [GFP]_high and [*vasa2*::mOr2]_high colocalizing to Vasa2+/[mOr2-Piwi1]_high cells (Fig. 5H-H′ and S8E′-E″; arrowheads). [GFP]_low and [*vasa2*::mOr2]_low proteins colocalize to cells of the parietal muscle region (Fig. 5I-I′ and Supplementary Fig. 8C-C′, E-E″; white arrows) and body wall epidermis (Supplementary Fig. 8E′-E″, I-I′; yellow arrows). Body wall and tentacle epidermis show relatively uniform [GFP]_low while [*vasa2*::mOr2]_low cells appear in patches (Fig. 5I-I′ and Supplementary Fig. 8A-B′, E-E″, I-I′), and are partly EdU+ (Supplementary Fig. 8A″). The absence of detectable Vasa2 protein (Supplementary Figs. 6A, C and 9J) or *mOr2* mRNA (Supplementary Fig. 7G, H) from body wall epidermis confirms that epidermal *vasa2*::mOr2 protein is likely not the result of endogenous *vasa2* or ectopic *mOr2* expression. Together, our results further support that both *piwi1P2A-GFP* and *vasa2*::mOr2 reporter lines highlight partly proliferative, somatic progeny cells derived from a rare population of mesenterial Vasa2+/[mOr2-Piwi1]_high cells (Figs. 5C, G and Supplementary Fig. 9B, K, L; Source Data).

Additionally, however, we also found cells labelled only by GFP or mOr2. [GFP]_low/*vasa2*::mOr2– cells located to the parietal and retractor muscle regions, and to the epidermis (Fig. 5I-I′, black arrowhead; Supplementary Fig. 8E-E′, H-H′, I-I′; white asterisks; Source Data). Conversely, GFP–/[*vasa2*::mOr2]_low cells are found in the

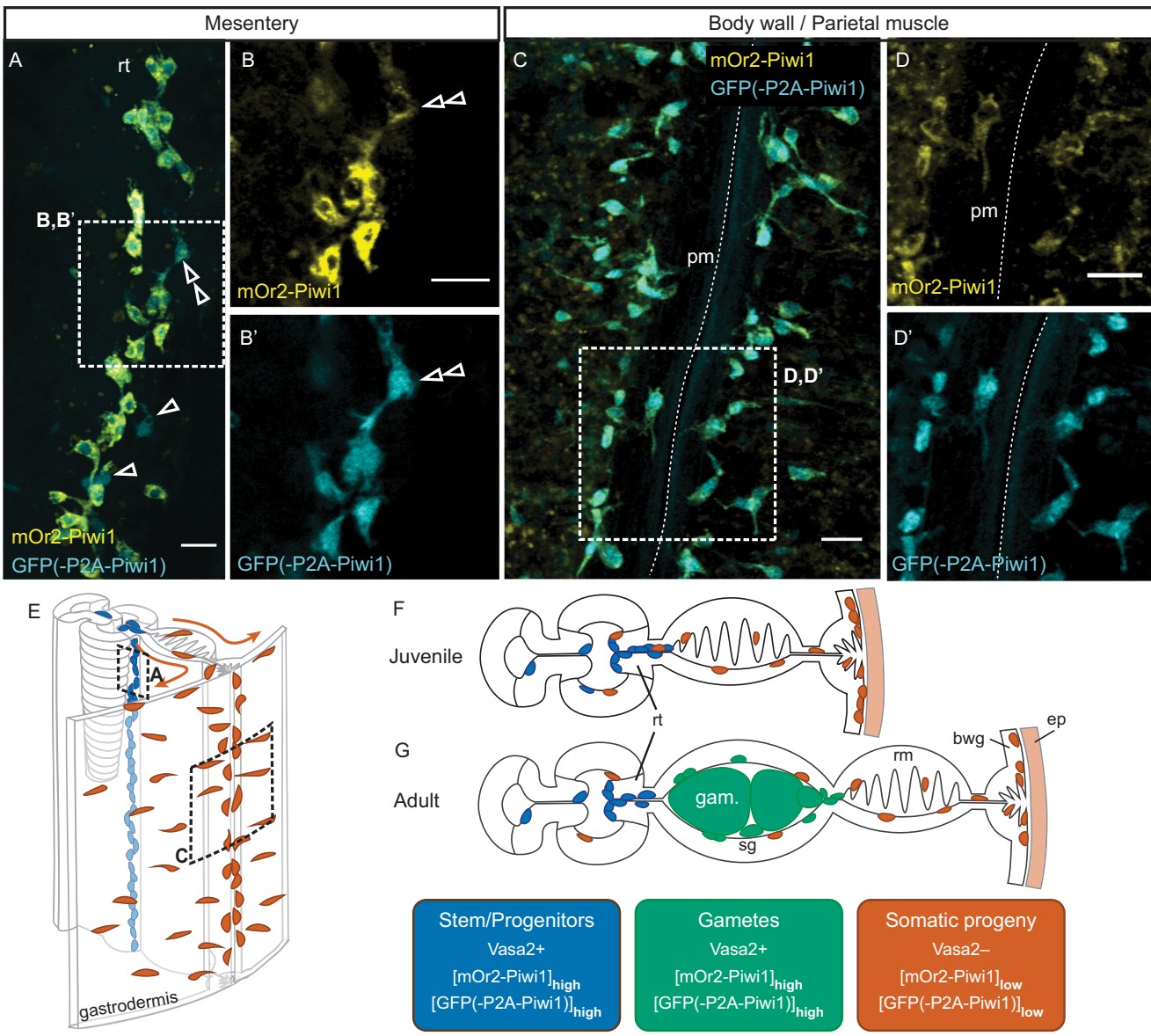

**Fig. 4 | The *piwi1*^mOr2/P2A-GFP^ reporter line suggests that the Vasa2+/Piwi1+ stem/progenitor cell population generates germ cells and putative somatic progeny. A–D'** Confocal images of whole-mount, double transgenic *piwi1*^mOr2/P2A-GFP^ juvenile polyps immunostained for mOr2 (**A–D**; yellow) and GFP (**A, B', C, D'**; cyan). As in adults, high levels of both reporter proteins were detected in basiepithelial cells along the reticular tract ('rt') of juvenile mesenteries (**A-B'**). Cells with [GFP]_low/[mOr2-Piwi1]_low localize along the parietal muscle and gastrodermal body wall of juveniles (**C-D'**). Cells with [GFP]_high/[mOr2-Piwi1]_low (**A-B'**; black double arrowheads) adjacent to [GFP]_high/[mOr2-Piwi1]_high cells indicates that unequal cytoplasmic inheritance of fluorophores occurred following a cell division. **E–G** Schematic 3-dimensional illustration of body column section (**E**) and cross-sections (**F, G**) summarizing the location and approximate levels of mOr2-Piwi1 and GFP(-P2A-Piwi1) proteins in the juvenile (**E, F**) and adult (**G**) mesentery. **E** Location of mesenterial Vasa2+/Piwi1+ stem/progenitor cells (blue) and their potentially migratory gastrodermal progeny cells (orange). bwg body wall gastrodermis, ep epidermis, m mesentery, n nucleus, o oocyte, pm parietal muscle, rm retractor muscle, rt reticular tract, sg somatic gonad. Scale bars: 10 μm (**A–D**). All experiments were performed twice with similar results.

cnidoglandular and ciliated tracts (Supplementary Fig. 8F-G', yellow asterisks; Source Data). These populations of mOr2-only or GFP-only cells could result from technical artefacts, such as differences in protein expression or degradation levels between GFP and mOr2, or from missing cis-regulatory elements in the 1,6kb-long *vasa2*::mOr2 promotor sequence. Alternatively, these cells could derive from an unidentified population of stem or progenitor cells that specifically express only Vasa2 or Piwi1, but for which we currently lack any further evidence (see discussion).

Overall, the large overlap between [GFP]_low and [*vasa2::mOr2*]_low somatic cells (Source Data) confirms that the potential progeny of the Vasa2+/[mOr2-Piwi1]_high cell population does not only include gametes, but also a diversity of somatic cells found throughout the polyp body column.

## Neuronal derivatives from Vasa2+/[mOr2-Piwi1]_high cell population

In juveniles, the parietal muscle region and adjacent body wall gastrodermis show a particularly high concentration of neural-like, putative progeny cells that are double-labelled in *piwi1*^P2A-GFP^/*vasa2::mOr2* animals (Supplementary Fig. 8C-C"). We therefore tested if the Vasa2+/[mOr2-Piwi1]_high cell population contributes to neuronal lineages by crossing *piwi1*^P2A-GFP^ and *vasa2::mOr2* lines with previously published neuronal reporter lines. Crossing the *vasa2*::mOr2 and *prdm14d*::GFP[42]

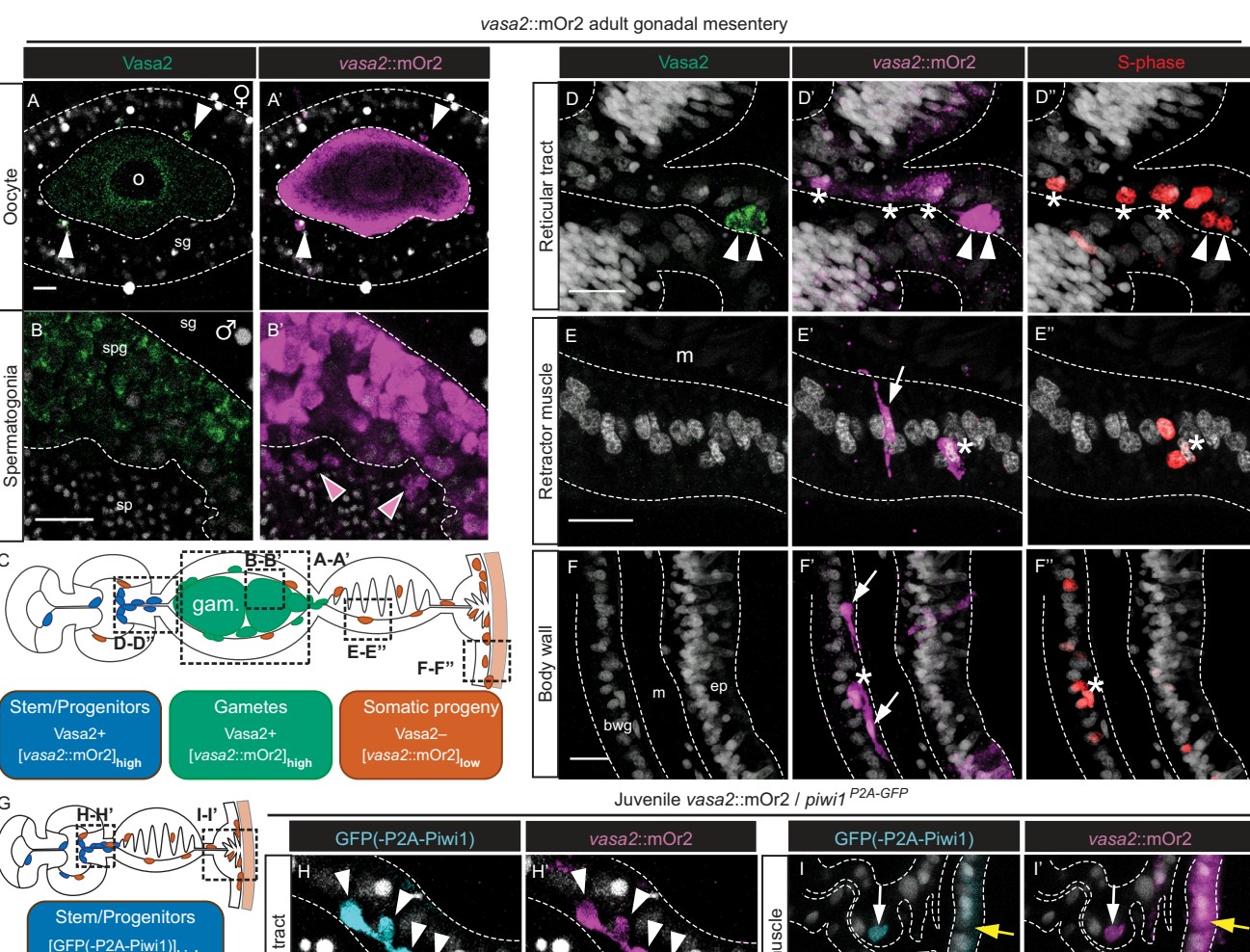

**Fig. 5 | *vasa2::mOr2* reporter line further indicates that both germinal and somatic cells derive from the Vasa2+/Piwi1+ stem/progenitor cell population.**
**A-B', D-F", H-I'** Confocal imaging stacks of cross-sectioned transgenic *vasa2::mOr2* adult mesenteries (**A-B', D-E**) or body wall (**F-F"**), and of *vasa2::mOr2/piwi1^P2A-GFP* juvenile mesenteries (**H-I'**) immunostained for Vasa2 (**A, B, D-F**; green), mOr2 (**A', B', D'-F', H', I'**; magenta), GFP (**H, I**; cyan) and S-phase nuclei (**D"-F"**; 3 days EdU pulse, red). Cross sections as indicated in (**C**) and (**G**). **A-F"** In adults, [*vasa2*::mOr2]high colocalizes with Vasa2 to oocytes (**A, A'**), spermatogonia (**B, B'**) and basiepithelial Vasa2+/EdU+ cells in the reticular tract (**D-D'**; arrowheads). In spermaries, mOr2 protein found in differentiating, Vasa2− spermatocytes likely resulted from cytoplasmatic inheritance (**B'**; magenta arrowheads). [*vasa2*::mOr2]low was detected in Vasa2−/EdU+ (asterisks) and Vasa2−/EdU− (arrows) cells in the reticular tract (**D-D"**), retractor muscle (**E-E"**) and body wall (**F-F"**). **C, G** Schematic summaries of the location and approximate levels of Vasa2, *vasa2*::mOr2 or GFP proteins in the adult (**C**) or juvenile (**G**) mesentery. (**H-I'**) [*vasa2*::mOr2]high and [GFP]high in juveniles colocalize to basiepithelial stem/progenitor cells of the reticular tract (**H-H'**; arrowheads). [*vasa2*::mOr2]low and [GFP]low colocalize to basiepithelial cells in the parietal muscle tract (**I-I'**; white arrows) and epidermis (yellow arrows). Note a single GFP + /*vasa2*::mOr2− cell in the parietal muscle tract (**I-I'**; black arrowhead). Grey: Hoechst DNA dye. bwg body wall gastrodermis, cgt cnidoglandular tract, ct ciliated tract, ep epidermis, it intermediate tract, m mesoglea, o oocyte, pm parietal muscle, rm retractor muscle, rt reticular tract, sg somatic gonad, sp sperm, spg spermatogonia. Scale bars: 10 μm (A-B, D-F) and 5μm (H-I). Experiments in (**A-F'**) performed twice with similar results. Experiments in (**H-I'**) performed once and results of both GFP(-P2A-Piwi1) and *vasa2*::-mOr2 were consistent with whole-mounts of the same double transgenic line (Fig. 4A-D') or in different transgenic combinations (Fig. 7A, C; Supplementary Fig. S9H', J').

---

lines shows that differentiated neurons along the parietal muscle, labelled by high *prdm14d*::GFP levels, were free of detectable *vasa2*::-mOr2 (Fig. 6A-A"; arrowhead). In contrast, early differentiation stages of [*prdm14d*::GFP]low neurons, characterized by short, putatively out-growing neurites, often contain [*vasa2*::mOr2]low protein (Fig. 6A-A"; arrows). Similarly shaped cells were also double-labeled after crossing *vasa2*::mOr2 with (i) *elav1*::cerulean that highlights a broad variety of differentiating neurons (Supplementary Fig. 10A-A")[43] or with (ii) *insm1*::GFP that labels neuroglandular progenitor cells and their progeny[18] (Supplementary Fig. 10B-B"). Together, co-labelling in early

differentiation stages of *elav1 +* , *insm1*+ or *prdm14*+ neurons suggests that their progenitors derive from Vasa2+/[mOr2-Piwi1]high cells. We further explored this possibility by crossing *piwi1^P2A-GFP* with the *soxB(2)::mOr2* line, which labels neural progenitors and their progeny in larvae and primary polyps[16]. In *piwi1^P2A-GFP*/*soxB(2)::mOr2* juveniles, GFP was not detected in differentiated *soxB(2)::mOr2*+ neurons (Fig. 6B-B"; arrowheads; inlets in Fig. 7B-B', D-D'). Also, no *soxB(2)::*-mOr2 was detected in Vasa2+/[mOr2-Piwi1]high cells (Fig. 7A-A'). Notably, however, cells scattered in the body wall gastrodermis and along parietal and retractor muscles were co-labelled by [GFP]low and

*soxB(2)*::mOr2 (Fig. 6B-B" and 7B-D'; arrows). Their EdU-labelled subset (Fig. 7D-D"; asterisks) likely corresponds to SoxB(2)+ progenitor cells. As putative *vasa2*::mOr2+ neural progenitor cells were previously co-labelled by *insm1::GFP*, we tested if *insm1::GFP* also overlaps with *soxB(2)*::mOr2 and indeed found multiple co-labelled cells (Supplementary Fig. 10C-C"), as previously shown at larval stages[18]. Together, our data strongly suggest that *soxB(2)*+ and *insm1*+ neural progenitor cells retain GFP and mOr2 proteins after cytoplasmatic inheritance from Vasa2+/[mOr2-Piwi1]$_{high}$ cells in juveniles. An alternative scenario, where *soxB(2)*+ and *insm1*+ neural progenitor cells develop independently of Vasa2+/[mOr2-Piwi1]$_{high}$ cells requires the expression of currently undetectable levels of both *piwi1* and *vasa2* genes (see also Fig. 6B and discussion). We therefore propose that the progeny of Vasa2+/[mOr2-Piwi1]$_{high}$ cells contributes to gastrodermal *soxB(2)*+ and *prdm14d*+ neural progenitors, *insm1*+ neuroglandular cell lineages and *elav1*+ neural lineages in addition to germ cells and other somatic cell types in the gastrodermis of *Nematostella* (Figs. 6C, 7E and 8).

## Discussion

Cnidarian stem cells have so far only been described in hydrozoans (e.g., *Hydra* or *Hydractinia*)[12,13]. Here, we used immunolabelling of Vasa2 and a mOr2-Piwi1 fusion protein generated by CRISP/Cas9-mediated knock-in to identify a spatially restricted population of proliferative, basiepithelial Vasa2+/[mOr2-Piwi1]$_{high}$ cells (termed 'Vasa2+/Piwi1 +' hereafter) in the mesenteries of the sea anemone *Nematostella vectensis*. These cells reside in the reticular tract of the septal filament and near the distal retractor muscle region in both juvenile and adult polyps, corresponding to the location previously described for potential PGCs[33]. The finding of basiepithelial Vasa2+/Piwi1+ cells at the basis of the septal filament of *Euphyllia ancora* and *Exaiptasia pallida* mesenteries suggests their evolutionary conservation among hexacorallians (i.e. sea anemones and stony corals)[32,33,44–46]. Quantification by flow cytometry showed that Vasa2+/Piwi1+ cells are rare in juveniles (~0.04%) or adult females (~0.4%) in *Nematostella*. Their ~10-fold higher abundance in adults is likely due to the presence of oogonia and small oocytes, which are absent in juveniles. The scarcity of Vasa2+/Piwi1+ cells could explain why recent single-cell RNA sequencing atlases from *Nematostella*[14,15,19,47], the soft coral Xenia[48] or the hard coral *Stylophora pistillata*[49] have so far failed to identify a non-gametal Vasa2+/Piwi1+ cell population. Their proliferation rate, as determined by EdU incorporation, was found to be distinctly lower in adults (~42%) than in juveniles (~73%) despite a 3x longer EdU pulse in adults. This likely reflects general differences in body growth rates between fast growing juveniles[50] (at 25 °C) and adults that present low asexual reproduction rates and relatively stable body sizes[51] at around 18 °C.

The occurrence of basiepithelial Vasa2+/Piwi1+ cells in juveniles and non-gonadal adult regions led us to explore if this cell population potentially contributes to somatic lineages in *Nematostella*. By characterizing a combination of a *piwi1*$^{mOr2}$, *piwi1*$^{P2A-GFP}$ and *vasa2*::mOr2 transgenic lines, we reveal the presence of reporter fluorophores in a diversity of somatic cell types in juvenile and adult polyps that lack any detectable *mOr2*, *piwi1* or *vasa2* mRNA or Vasa2 protein (Fig. 8A). In both juveniles and adults, cells with [mOr2-Piwi1]$_{low}$ are abundant (~53% or ~20%, respectively) and in large part proliferative (~63% or ~41%, respectively). Using confocal imaging, we show that many proliferative cells with [*vasa2*::mOr2]$_{low}$ or [GFP(-P2A-Piwi1)]$_{low}$ in the gastrodermis also exhibit polarized shapes and basiepithelial locations. Together, our data thus suggests that the Vasa2+/Piwi1+ cell population generates transit-amplifying progenitor cells that migrate towards or into the body wall (Figs. 6C and 8). This scenario is supported by a previous report showing that EdU-labelled cells from transplanted mesenteries migrated into the non-labelled body wall[52].

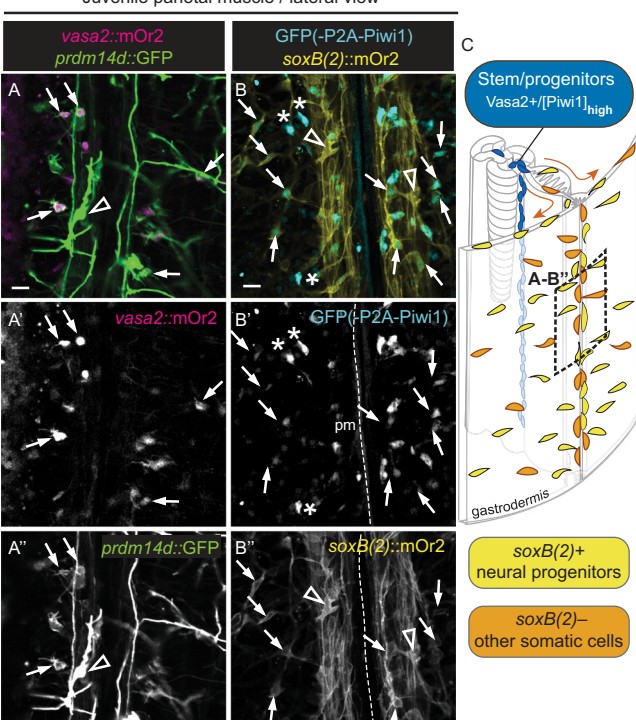

**Fig. 6 | Some developing neuronal cells in the gastrodermis are labelled by *vasa2*::mOr2 or *piwi1*$^{P2A-GFP}$.** Confocal live imaging of the parietal muscle region from *vasa2*::mOr2/*prdm14d*::GFP (**A-A"**) or *piwi1*$^{P2A-GFP}$/*soxB(2)*::mOr2 (**B-B"**) double transgenic reporter lines at the mid-body level of juvenile polyps (see **C**). **A-A"** Differentiated gastrodermal neurons in the parietal muscle express [*prdm14d*::GFP]$_{high}$ but no *vasa2*::mOr2 protein (black arrowheads). In contrast, [*prdm14d*::GFP]$_{low}$ colocalizes with [*vasa2*::mOr2]$_{low}$ in cells along the parietal muscle and in the body wall gastrodermis (arrows). **B-B"** Similarly, neuronal cell bodies within axonal bundles along the parietal muscle show high levels of the neural progenitor marker gene *soxB(2)*::mOr2 but no GFP protein (black arrowheads). While some [GFP]$_{high}$ cells show no *soxB(2)*::mOr2 (asterisks), relatively low levels of *soxB(2)*::mOr2 are detected in a subset of [GFP]$_{low}$ cells with neuron-like shape (arrows). **C** 3-dimensional illustration of a juvenile body column section highlighting the gastrodermal location of neural (yellow) among other somatic progeny (orange) that potentially derive from mesenterial Vasa2+/Piwi1+ stem/progenitor cells (blue). Arrows indicate putative migration of transit-amplifying progenitors. pm parietal muscle. Scale bars: 10μm. Experiments performed twice with similar results.

Neurite-like extensions on some progeny cells in the body wall gastrodermis or along the muscle tracts alluded to a potential neuronal contribution from the population of Vasa2+/Piwi1+ cells. Previous studies using transgenic reporter lines and single-cell RNA-seq analysis have shown that multipotent *soxB(2)*+ neural progenitor cells generate gland cells, neurons and cnidocytes during larval and potentially also post-larval development[14,16,18]. So far, however, the cell populations upstream of *soxB(2)*+ neural progenitors have remained unidentified. Here, we crossed *piwi1*$^{P2A-GFP}$ or *vasa2*::mOr2 with neuronal reporter lines to show a partial overlap between [GFP]$_{low}$ or [*vasa2*::mOr2]$_{low}$ cells and reporter fluorophores labelling neural progenitors (*soxB(2)*, *prdm14d*), neuroglandular progenitors (*insm-1*) or early differentiating neurons (*elav1*). We therefore propose that the population of basiepithelial Vasa2+/Piwi1+ cells contribute to at least a subset of *soxB(2)*+ and *prdm14d*+ neural progenitors, *insm1*+ neuroglandular lineages and *elav1*+ neural lineages in the juvenile and adult gastrodermis (Figs. 6C, 7E and 8). Interestingly, a recent publication similarly shows that a

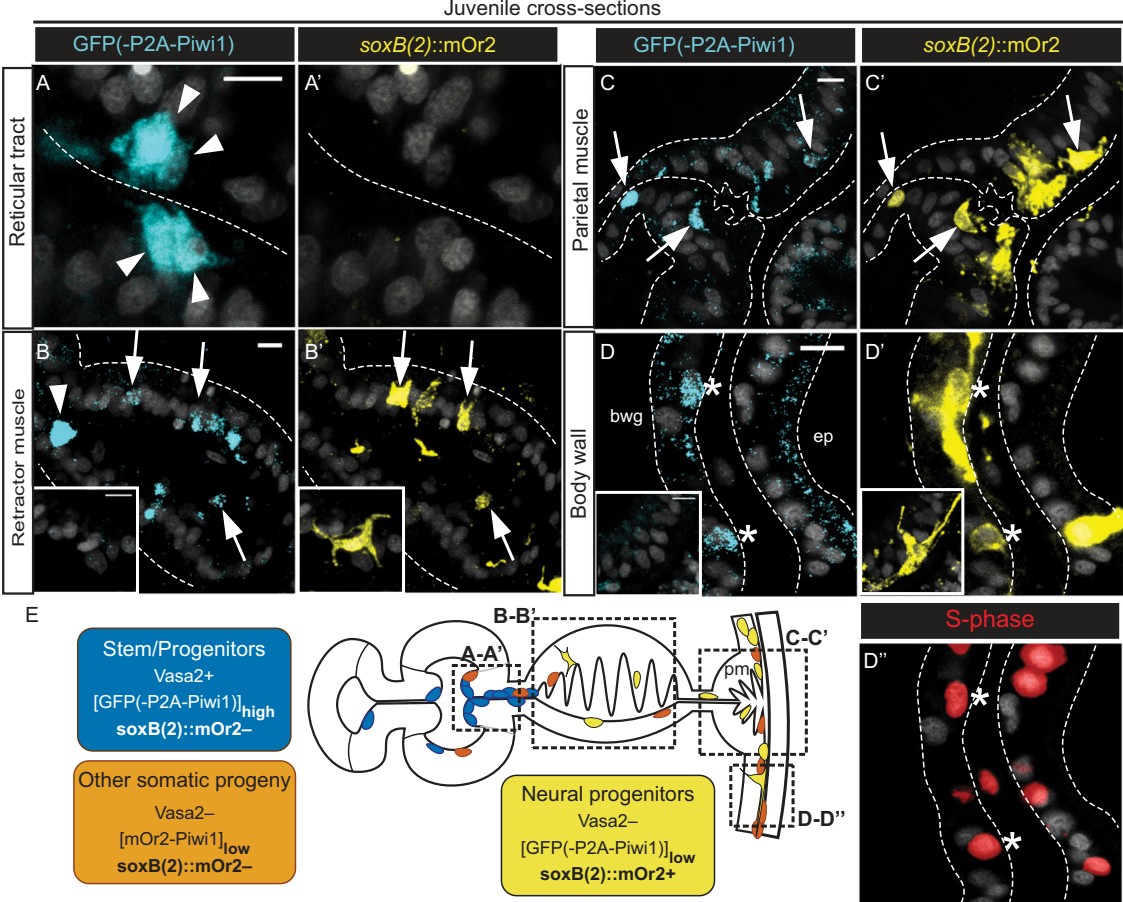

**Fig. 7 | Gastrodermal neurons potentially derive from the Vasa2+/Piwi1+ stem/progenitor cell population. A–D'** Confocal imaging stacks of cross-sectioned *piwi1^P2A-GFP*/soxB*(2)*::*mOr2* juvenile mesenteries immunostained for GFP (**A-D**; cyan), mOr2 (**A'-D'**; yellow) and EdU-labelled for S-phase detection (**D''**; 1 h pulse, red). Reticular tract stem/progenitor cells with [GFP]high do not express *soxB(2)*::mOr2 (**A, A'**, arrowheads), while neurons with high levels of *soxB(2)*::mOr2 are GFP– (inlets in **B, B', D, D'**). [GFP]low/[*soxB(2)*::mOr2]low cells localize to mostly basiepithelial cells in the retractor muscle (**B-B'**; arrows), parietal muscle (**C-C'**; arrows) and body

wall gastrodermis (**D-D''**; asterisks) and are partially EdU+ (**D''**; asterisks). **E** Schematics depicting the location and approximate levels of Vasa2, GFP(-P2A-Piwi1) and *soxB(2)*::mOr2 in stem/progenitor cells (blue) and their potentially derived neural progenitors (yellow) and other somatic cells (orange). Grey: Hoechst DNA dye. bwg body wall gastrodermis, ep epidermis, pm parietal muscle. Scale bars: 5 μm (**A-D''**). Experiments performed once but consistent with replicated results of independent in vivo imaging experiments (Fig. 6B-B'') and GFP(P2A-Piwi1) detection (see for example Fig. 4).

*piwi1*::mOr2 transgenic line labels germ cells, epidermis and neuronal cell populations[19]. In addition, *piwi1*::mOr2 also labels gland cells (observed only in our *vasa2*::mOr2 line) and cnidocytes[19]. These differences are likely methodological due to higher, mOr2 fluorophore expression levels from integrating transgene concatemers by I-SceI meganuclease-mediated transgenesis[53]. As in our *vasa2*::mOr2 reporter line, *piwi1*::mOr2 could be cytoplasmically inherited over more cell divisions, resulting in a more complete cell lineage labelling than in single-copy, knock-in lines. This hints to the possibility that Vasa2+/Piwi1+ cells also contribute to cnidocyte and gland cell types lineages (Fig. 8B).

Notably, despite the absence of detectable endogenous *piwi1*, *vasa2* mRNA or Vasa2 protein, the epidermis shows expression of GFP, mOr2 or mOr2-Piwi1 proteins in all three independently generated reporter lines reported here and in a recently published *piwi1::mOr2* line[14]. We also did not detect *mOr2* mRNA in the epidermis of *vasa2::mOr2* or *piwi1^mOr2* juveniles, showing that epidermal mOr2 protein is unlikely to originate from endogenous or ectopic *mOr2* expression in the epidermis. In addition, neither *vasa2* nor *piwi1* genes were identified as markers for epidermal cell clusters in recent single-cell RNA-seq datasets[14,47]. Our data is therefore best explained by the assumption that the juvenile epidermis derives from the population of

mesenterial Vasa2+/Piwi1+ cells. Alternative explanations, such as the presence of an independent population of Piwi1+ and/or Vasa2+ epidermal stem cells, as found in *Hydra*[54,55], would imply that expression levels of both *vasa2* and *piwi1* are below detection limits by current technical means.

Altogether, our results suggest that basiepithelial Vasa2+/Piwi1+ cells in the mesenteries of juvenile and adult polyps constitute a multipotent population of stem and/or progenitor cells with potential to generate both gametes and somatic cell populations (Fig. 8B). Currently, however, we can only speculate about the stemness or potency of individual Vasa2+/Piwi1+ cells and thus their homo- or heterogeneity within the cell population (Fig. 8B). Nevertheless, our data is not compatible with previous assumptions that the Vasa2+/Piwi1+ cells forming in the mesentery during late larval development are PGCs, i.e. uniquely deriving germline[33,56]. Instead, the population of Vasa2+/Piwi1+ cells could consist of (i) a homogenous population of multipotent stem/progenitor cells without soma-germline separation, (ii) a heterogenous population of stem/progenitor cells with multipotent somatic cells segregated from primordial germ cells, or (iii) a highly heterogenous pool of segregated uni-/multipotent somatic and germline lineages (Fig. 8B). The first model is supported by recent genomic data from stony corals, which are closely related to sea

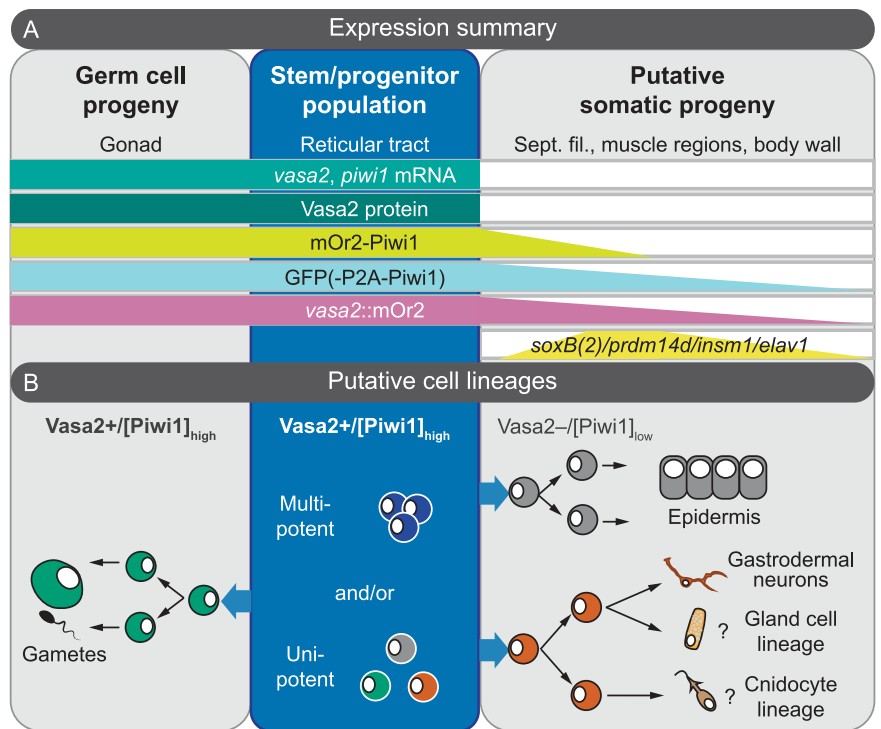

**Fig. 8 | Summary of gene and protein expression (A) and model of putative cell lineages deriving from the Vasa2+/Piwi1+ cell population (B). A** Schematic summary of the location of *vasa2* and *piwi1* mRNA expression and the relative levels and locations of Vasa2, mOr2-Piwi1, *vasa2*::mOr2 and GFP(-P2A-Piwi1) proteins of different reporter lines in Vasa2+/Piwi1+ stem/progenitor (blue box), germ and potential somatic progeny cells. Fluorophores driven by *sox(B)2*, *insm*, *prdm14* and *elav1* partially co-localize with mOr2-Piwi1, *vasa2*::mOr2 and GFP(-P2A-Piwi1) in neuronal and possibly neuroglandular lineages. Based on our work, we hypothesize that all Vasa2 protein is asymmetrically retained in Vasa2+/Piwi1+ stem/progenitor cells and becomes undetectable in any somatic progeny cells. See also Source Data

file for a detailed summary of gene and protein expression locations. **B** Hypothetical cell lineage relationships between an adult Vasa2+/Piwi1+ multi-potent stem/progenitor cell population and their germinal and somatic progeny. The homo- or heterogeneity of the Vasa2+/Piwi1+ cell population, the potency of their individual cells and the precise lineages of their somatic progeny remain to be further investigated. Our data is however inconsistent with a model where post-larval Vasa2+/Piwi1+ cells consist solely of primordial germ cells as previously suggested[33,56]. Instead, our data suggests a model where the Vasa2+/Piwi1+ cell population not only generates germ cells, but also somatic progeny.

anemones, demonstrating a common genetic lineage of gametes and somatic cells throughout adulthood[23,24]. Future studies, for example by single cell transplantation or the use of photoconversion for single cell lineage tracing, are necessary to reveal the stemness and potency of individual Vasa2+/Piwi1+ cells in *Nematostella*.

The population of Vasa2+/Piwi1+ cells described here shows strong similarities to the i-cell population, so far considered hydrozoan-specific. Both share a high nucleus-to-cytoplasm ratio, a basiepithelial location and the expression of conserved GMP genes (e.g., *vasa* and *piwi*)[5,30,35,36]. In addition, both populations generate germline, neurons and potentially glandular and cnidocyte lineages[13,19,55,57]. As GFP and mOr2 remain only transiently present in putative progeny cells, our proposed cell lineage reconstructions by cytoplasmatic inheritance of fluorophores are likely incomplete. It thus remains unclear if the Vasa2+/Piwi1+ cell population contributes to all or only a subset of *soxB(2)* lineages in juvenile and adult *Nematostella* polyps.

ASCs holding germinal and somatic potential (i.e. PriSCs) have been consistently found in animals displaying high body plasticity and regenerative abilities, such as sponges (archeocytes and choanocytes), hydrozoans (i-cells), acoels and planarians (neoblasts)[8,10,11,13,58,59]. Here, we report the first population of Vasa2+/Piwi1+ cells as candidates for adult stem cells in a sea anemone. Their discovery in *Nematostella* also sets the basis for uncovering the cellular and molecular mechanisms that underlie body plasticity in sea anemones[50] and corals, and for further elucidating the evolution of animal stem cell systems. Our finding also has the potential to spark the exploration of stem cell biology in corals that could inform the development of stem cell-based

approaches for supporting conservational efforts for engineering and repopulating heat-resistant coral reefs[60].

## Methods

### Animal culture

*Nematostella vectensis* polyps derive from CH6 females and CH2 males constituting the original culture[61]. Female and male adult animals are maintained in separate boxes in the dark at 18 °C in 1/3 filtered sea water (*Nematostella* medium, NM), daily fed with fresh *Artemia nauplii*. Spawning is induced approx. every 3 weeks by a light and temperature shift (25 °C) overnight (12 h), as previously described[62]. Embryos are raised at 25 °C, and daily feeding with *Artemia nauplii* is incorporated 10 days post-fertilization, approximately. For all adult data, the sex is indicated. The sex cannot be determined in juveniles, and thus all data shown is from a mix of males and females.

### I-Sce I meganuclease-mediated generation of a *vasa2* reporter line

A 1.6kb-long genomic fragment upstream of the putative *vasa2* transcription start site was selected taking into account the genome-wide histone marks H3K27ac, H3K4me1, H3K4me2, H3K4me3 and ChIP-Seq data for P300 (transcriptional coactivator) in adult females[63]. The genomic fragment that included the regulatory elements was successfully cloned into an existing pJet1.2 plasmid backbone (Thermo-Fisher) containing an mOrange2 (mOr2) ORF sequence and SceI sites. The resulting vector consisted of 1.6kb *vasa2* promoter upstream of mOr2, altogether flanked by inverted I-Sce I sites for I-Sce I mega-nuclease-mediated integration into the genome. Microinjection was

performed as previously described[39]. 150 injected zygotes were raised until juvenile stage, then selected if presenting gastrodermal mOr2 signal. At adult stage one single animal displaying mOr2 in its developing gametes was selected as a founder and outcrossed with wild type to establish the F1 generation.

### CRISPR-Cas9 mediated generation of *piwi1* knock-in lines

A previously published protocol for the generation of CRISPR/Cas9-mediated knock-in lines in *Nematostella*[64] was used and adapted as follows. Two guide RNA (gRNA) target regions with putative cutting sites located 174bp upstream or 229bp downstream of the start codon of the *Nematostella piwi1* (v1g79423) were designed using CRISPOR[65]. Templates for gRNAs were generated using annealed and PCR-amplified oligos[66]: two T7-and gRNA-encoding oligos (ThermoFisher, desalted): 5′-GAAATTAATACGACTCACTATAGacaccacggttccaatatggG TTTTAGAGCTAGAAATAGCAAG-3′ (upstream of stop codon); 5′-GAAATTAATACGACTCACTATAGGATGTCAGTGGAACCACGTGG TTT TAGAGCTAGAAATAGCAAG-3′ (downstream of stop codon); invariant reverse primer (ThermoFisher, desalted): 5′-AAAAGCACCGACTCG GTGCCACTTTTTCA AGTTGATAACGGACTAGCCTTATTTTAACTTGCT ATTTCTAGCTCTAAAAC-3′.

Guide RNAs were in vitro transcribed using a T7 MegaScript transcription kit (ThermoFisher) followed by ammonium chloride precipitation and diluted in nuclease-free $H_2O$ to a final concentration of 1,5μg/μl. The DNA donor fragment for homology-mediated repair (HDR) consisted of the coding sequences for the mOr2 protein followed by a GGGGS[2] linker (mOr2-Piwi1) or a GFP protein followed by a P2A sequence (GFP-P2A-Piwi1) cloned in frame to the 5′ -end of the *piwi1* gene open-reading frame. In addition, the construct included a 995bp long 'left' homology arm (upstream of Piwi1 coding region) and a 1184bp 'right' homology arm (downstream and overlapping the Piwi1 coding region). The entire construct was cloned into a pJet1.2 plasmid backbone (ThermoFisher) using Gibson assembly master mix (NEB)[67]. Donor DNA fragment was PCR-amplified using 5′-biotin-labelled oligos flanking the donor fragment to increase integration efficiency as previously described for medaka fish[68]. Oligo sequences (ThermoFisher; desalted) are: Forward oligo 5′-[Biotin]-TACGACTCACTATAGG GAG AGCGGC-3′; reverse oligo 5′-CCATGGCAGCTGAGAATATTGTAGGA-3′.

The Cas9-mediated knock-in injection was performed by modifying previously described protocols for CRISPR/Cas9-mediated mutagenesis and using 0.75μg/μl nls-Cas9 protein (PacBio), 75ng/μl of each guide RNA, 70ng/μl column-purified donor DNA, and modified injection buffer containing 220mM KCl[69,70]. After injection of ~1600 zygotes and subsequent raising, juvenile polyps displaying mOr2 or GFP signal in their gastrodermis were selected and raised until sexual maturity. At the adult stage, polyps presenting mOr2 or GFP signal in the developing gametes were selected as founder animals and outcrossed with wild type animals to generate F1 generation animals. The successful integration of the GFP-P2A-Piwi1 fragment knock-in was validated on the transcript level by in vitro synthesis of cDNA based on extracted total RNA from a mix of 3 days-old larvae resulting from a cross of two heterozygous F1 animals. Based on this cDNA, the transitions between the 5′ UTR and GFP-encoding region, and between the P2A peptide-encoding region and the 5′ end of the Piwi1 open-reading frame have been amplified using oligos outside of the 'homology arm' regions, if possible (exception: short Piwi1 5′UTR was fully included within homology arm). Sanger sequencing of gel-eluted and column-purified PCR fragments confirmed the flawless integration of the GFP-P2A fragment directly upstream and in frame with the Piwi1 open-reading frame. Oligos used for PCR & sequencing of GFP-P2A-Piwi1: Piwi1-T-Forward-1: 5′-GAGAGACAGAGCGTTTGAGAGAGAGAC-3′; Piwi1-T-Forward-2: 5′-GAGTTGTGTAATTTTAGGTAAGTTTT GGTC-3′; Piwi1-T-Reverse-1: 5′-TTCCTTCCCACTTGCTTCATGTC-3′; Piwi1-T-Reverse-2: 5′-GTAACCTGGCCACAGCTCAAGC-3′; GFP-Forward (used with Piwi1-Reverse-1 or -2): 5′-TCCGCCCTGAGCAAAGACC-3′; GFP-Reverse (used

with Piwi1-Forward-1 or -2): 5′-TTGCCGTAGGTGGCATCGC-3′. The successful and flawless integration of the mOr2-Piwi1 knock-in was validated on the genomic level by extracting genomic DNA from mixed tentacle clips of several homozygous F2 animals from the same founder F1 animals. Based on this gDNA, the transitions between the 5′UTR of the *piwi1* gene and mOr2, and between the GGGGS[2] linker and 5′ end of the Piwi1 ORF have been amplified using oligos outside of the 'homology arm' regions. Gel elution and Sanger sequencing confirmed the flawless integration. Oligos used: Piwi1-G-Forward: CGCGATACA-CACTAAACATCTAGGC; Piwi1-G-Reverse: GCAAA AGATAGGACAATCA GCCCCATC; mOrange2-Reverse (used with Piwi1-G-Forward): CTGTCTGAAAGCCCTCGTATGGTC; mOrange2-Forward (used with Piwi1-G-Reverse): CAAGGCAAAGAAGCCAGTGCAGC.

### Gene cloning and RNA probe synthesis

A set of *Nematostella* GMP gene orthologs was based on a previous publication[10] and comprised *piwi1*, *piwi2*, *vasa1*, *vasa2*, *pl10* and *tudor*. RNA of whole adult female *Nematostella* was extracted using Trizol (ThermoFisher) as indicated by the manufacturer. cDNA was synthetized using the SuperScript III First-Strand Synthesis System (ThermoFisher), followed by PCR amplification of fragments of the genes of interest. Primers designed with Primer3Plus (http://www.bioinformatics.nl/cgi-bin/primer3plus/primer3plus.cgi). Cloned gene fragments were inserted into the pGEM-T Easy vector (Promega, A1360) and transformed into One Shot Top 10 chemically competent *E. coli* (Invitrogen). Sanger sequencing verified the cloned sequences at the sequencing facility of the Department of Biological Sciences, University of Bergen, Bergen, Norway. Digoxygenin (DIG)-labelled antisense riboprobes were generated using a T7 or SP6 MEGAscript Kit (Invitrogen, AMB1334/AMB1330) and DIG RNA Labelling Mix (Roche) as previously described[71]. Oligos used for PCR: *piwi1* and *piwi2* as in ref. [32]; *vasa2* and *tudor* as in ref. [33]; *vasa1*-Forward: CCCAAAC-CAAGCCAACCAAGGC, *vasa1*-Reverse: ATGTCAAGGC CACGAGCAGC; *pl10*-Forward: CTTAGCAGGATCTACATGGAAGGGC; *pl10*-Reverse: CCAGTCCTGACCACCGCTG ; mOr2-Forward: ATGGTGAGCAAGGGC GA GG; mOr2-Reverse: GTTCCACGATGGTGTAGTCCTCG.

### Colorimetric in situ hybridization

Adult female polyps kept in NM were left to relax in petri dishes, then $MgCl_2$ was carefully added and mixed in until reaching a 0.1 M $MgCl_2$/NM solution. Animals were left to relax for 30 min. The body cavity of the animals was flushed with this medium through the mouth opening to ensure full extension of the body column and mesenteries. The polyps were then transferred to a new dish containing 3.7% Formaldehyde NM solution, and head and physa were cut off with a scalpel. The body column was opened longitudinally with microdissection scissors to ensure penetrance of the fixative. The tissue was gently transferred into 1xPBS/3.7% Formaldehyde/0.5% DMSO/0.1% Tween20 to fix overnight at 4 °C. Afterwards, mesenteries with or without body wall were dissected in fixative and cut in ~3–5 mm long pieces. The tissue pieces were washed in 1x PBS/0.1% Tween20, followed by several washes in 100% methanol until the brown pigment was completely removed. Tissue pieces were finally stored in 100% methanol at −20 °C.

The urea-based in situ hybridization protocol was adapted from a previous study[72], with a series of changes as previously described[64]. After progressive rehydration in MeOH/PTx (0.3% Triton X-100 in 1xPBS pH 7.4), the tissue was digested with Proteinase K 2.5 µg/ml for 5 min at room temperature (RT). An additional fixation step (2 min in 0.2% glutaraldehyde/PTx followed by 1 h in 3.7% formaldehyde/PTx) was added before overnight blocking in a hybridization mix consisting of 50% 8M urea, 5x SSC pH 4.5, 0.3% Triton X-100, 1% SDS, 100 µg/ml heparin and 5 mg/ml Torula yeast RNA. Background staining could be reduced by adding 5% dextrane sulfate (MW > 500,000, Sigma-Aldrich) and 3% Blocking Reagent (Roche) to the hybridization mix

during overnight blocking and hybridization of the probe, and hybridization occurred over 2 days. The probe concentration was 0.75 ng/μl. Stringent washes varied between 2x SSC/0.3% Triton X-100 (SSCT) and 0.1x SSCT. After the SCCT washes, the tissue pieces were washed with 1x PBS/0.1%BSA/0.3% Triton X-100. Pieces of tissue were incubated in 500 ml of NBT/BCIP solution (4.5 μl/ml NBT, 3.5 μl/ml BCIP in Alkaline Phosphatase buffer) (Roche 11383213001 and 11383221001) in the dark from some minutes to several hours, depending on the signal. Once stained, the pieces of tissue were quickly washed in 100% ethanol for 3 min, then twice in PBTx and finally stored in 80% glycerol at 4 °C.

Before proceeding with the vibratome sectioning overview pictures of the stained pieces of tissue embedded in glycerol were taken in the microscope Leica M165 FC with the Leica DFC450 C camera using the Leica Application Suite X (LAS X) software.

## Immunofluorescence and S-phase labelling on whole-mount tissues

Wild-type and F1 transgenic reporter animals at adult and juvenile (~45 days after fertilization) stages were selected and fixed 24 h after the last feeding event. To label cells in S-phase, animals were placed in freshly prepared 100 μM EdU (Invitrogen) in 2%DMSO/NM before fixation. Incubation time was 1 day for juveniles, and 3 days for adults (medium was replaced daily). Animals were relaxed using $MgCl_2$, then fixed in 3.7% Formaldehyde NM for 1 h at room temperature and dissected in this same solution in a petri dish. Fixative was washed thoroughly in 1x PBS/0.2% Tween20, followed by a series of dehydration washes (20–50–100% methanol in 1x PBS/0.2% Tween20). 100% methanol washes were applied until pigment was completely removed from the tissue. Samples were stored in 100% methanol at -20 °C. After progressive rehydration of the tissue in 1xPBS/0.2% Triton X-100, for samples incubated in EdU a Click-it reaction was performed using the reagents of a Click-iT™ EdU Cell Proliferation Kit for Imaging (Invitrogen), following the manufacturer's protocol. After a 30 min incubation in the Click-it staining reaction, pieces were washed in in 1xPBS/0.2% Triton X-100.

For the immunofluorescence, tissue pieces were blocked in 1x PBS/10% DMSO/5% normal goat serum (NGS)/0.2% Triton X-100 for 2 h at room temperature. Primary antibody incubation was performed in 0.1% DMSO/5% NGS/0.2% Triton X-100 overnight at 4 °C using the following antibodies: rabbit anti-Vasa2 1:1000[32], mouse anti-Vasa2 1:500[33], rabbit anti-DsRed 1:100 (Takara Bio Clontech 632496), mouse anti-mCherry 1:100 (Takara Bio Clontech 632543), mouse anti-GFP 1:250 (Abcam Ab1218), rat anti-α-Tubulin (YL1/2 clone) 1:100 (Abcam Ab6160) and FluoTag®-X4 anti-GFP (N0304). After washes in 1xPBS/0.2% Triton X-100, tissue was blocked in 1xPBS/5%NGS/0.2% Triton X-100 for 30 min at room temperature. Hoechst or DAPI nuclear staining (ThermoFisher) and secondary antibody incubation was performed in 1x PBS/5% NGS/0.2% Triton X-100 overnight at 4 °C using the following antibodies: goat-anti-mouse-Alexa488/568 (LifeTech A11001, A11004), goat-anti-rabbit-DyLight488 (LifeTech 35552), goat-anti-rabbit-Alexa568/647 (LifeTech A11011, A21244) and goat-anti-rat-Alexa633 (LifeTech A21094). Finally, tissue pieces were washed thoroughly in 1xPBS/0.2% Triton X-100 and stored at 4 °C until sectioning.

## Vibratome sectioning

Vibratome sectioning was performed as previously published[73], with a series of changes. Stained pieces of mesenteries were selected for embedding in gelatin-albumin medium prepared in advance (22.5 ml PBS (10x), 1.1 g of gelatin Type A Sigma G1890, 67.5 g of albumin Sigma A3912 and 225 ml H20). Pieces were transferred into a plastic mold containing a freshly prepared mixture of 1/4 37%FA gelatin-albumin medium and let solidifying O/N at 4 °C in a sealed humid chamber. The solidified gelatin blocks were removed from the mold and placed in PTw to avoid drying. Sectioning was performed with the vibratome Leica VT1000 S, speed = 4, frequency = 6, and section size 20 μm. Slices

were stored in PTw in a well plate at 4 °C until subsequent mounting and imaging.

## Transmitted light and confocal imaging

In situ hybridization and immunofluorescence gelatin slices were mounted on slides (Electron Microscopy Sciences 63418-11) with glycerol and sealed with coverslip (Menzel-Gläser 18 × 18mm) and clear nail polish. Transmitted light pictures of ISH cross sections were taken in the microscope Nikon Eclipse E800 (60x oil) with the Nikon Digital Sight DS-U3 camera, NIS-Elements software. Immunofluorescence whole mount tissue pieces and cross sections were imaged either on a Leica SP5 confocal microscope (standard PMT detectors, 20x/40x/63x oil-immersion objectives), or an Olympus FV3000 confocal (standard PMT detectors, 40x/63x silicon-immersion objectives). Live juvenile polyps relaxed in $MgCl_2$ were mounted and imaged on the Olympus FV3000 confocal. Transmitted light images and confocal stacks were processed, cropped and adjusted for levels and colour balance with Fiji.

## Trypsin/formaldehyde-based cell dissociation and fixation

Juveniles were dissociated in pools of 12 polyps while adult females were dissociated as individuals. The resulting cell suspensions were treated as a biological replicates. Cell dissociation was carried out using a previously published protocol[50]. Briefly, after relaxation, animals were washed with calcium- and magnesium-free *Nematostella* medium (CMF/NM) followed by CMF/NM containing 0.195% ethylenediaminetetraacetic acid (CMF/NM+E). Animals were then incubated for 5 min at 37 °C in preheated CMF/NM+E containing 0.25% Trypsin (w/v). Homogenization was performed by pipetting and trypsinization was stopped by adding cold CMF/NM containing 1% BSA and 2.5% of Foetal Bovine Serum. Cells were centrifuged for 5 min at 800 g and at 4 °C, resuspended in 1x PBS/1% BSA, filtered through a pre-wetted 50 μm CellTrics strainer (Sysmex) and fixed with 3.7% formaldehyde for 30 min at RT in the dark. Finally, the cell suspension was centrifuged at 800 g for 5 min at 4 °C and was washed twice with 1x PBS/1% BSA. The final cell pellet was then resuspended in 90% Methanol/0.1x PBS/1% BSA and stored at −20 °C.

## S-phase labelling and immunofluorescence on cell suspensions

Before dissociation, juvenile or adult female polyps were incubated for 1 day (juveniles) or 3 days in 100 μM EdU in 2% DMSO/NM or in 2% DMSO/NM (as control), as for whole mount imaging. After cell dissociation (see above), cells were rehydrated with two washes of 1x PBS/1% BSA after centrifugation at 800 g for 5 min at 4 °C. After pelleting, cells were then permeabilized with 0.2% Triton X-100 in PBS for 15 min at RT and washed with 1X PBS. Then, cells were incubated in Click-it reaction cocktail containing Alexa488 fluorophore azide for 30 min at RT, following the manufacturer's instructions (ThermoFisher, C10337). Cells were then washed twice with 0.2% Triton X-100 in PBS after centrifugation at 800 g for 5 min at 4 °C. The immunofluorescence protocol for cell suspensions was adapted from a previously published protocol for whole-mount tissues[33]. Following the EdU Click-it reaction, cells were blocked in 1x PBS/10% DMSO/5% NGS/0.2% Triton X-100 for 30 min at RT. Primary antibody incubation was in 1x PBS/0.1% DMSO/5% NGS/0.2% Triton X-100 overnight at 4 °C in the dark. The primary antibody used was rabbit anti-DsRed 1:500 (Takara Bio Clontech 632496). After two washes with 1x PBS/0.2% Triton X-100 and centrifugations at 800 g for 5 min at 4 °C, cell suspensions were incubated with the secondary antibody goat-anti-rabbit-Alexa568 1:500 (LifeTech, A11011) in 1x PBS/0.1% DMSO/5% NGS/0.2% Triton X-100 for 30 min at RT in the dark. Negative controls were stained only with the secondary antibody. Finally, cells were washed twice with 1x PBS/0.2% Triton X-100 after centrifugation at 800 g for 5 min at 4 °C and resuspended in 1x PBS/1% BSA containing FXCycle Violet 1 μg/ml (ThermoFisher,

F10347) for DNA staining. Cells were stored at 4 °C and not washed before performing flow cytometry within 24 h.

## Flow cytometry analysis

Flow cytometry was performed on a BD LSRFortessa (BD Life Sciences) instrument with 407 nm, 488 nm, 561 nm and 640 nm lasers, and the resulting data was analyzed using FlowJoV10.9 (BD Life Sciences). Graphical representation of the gating strategy is visualized in Supplementary Fig. 5. Briefly, debris was excluded based on size and granularity in the FSC-A/SSC-A gate, cell doublets based on FSC-A/FSC-H parameters, and high complexity events based on and FSC-A/SSC-W parameters. DNA dye intensity in area over height on the linear scale was used to select the pool of cells from which the fractions of mOr-Piwi1+ and EdU+ cells were analyzed, by sub-gating on Piwi1-mOr and EdU labeling fluorescence with reference to negative controls, respectively.

## Data visualization

Box plots show proportions of cell populations and EdU index. The centre bar shows the median and the bounds represent the 25th and 75th percentiles, respectively. Whiskers extend to the maxima within 1.5x the interquartile range above the upper quartile and to the minima within 1.5x the interquartile range below the lower quartile. Black dots represent all data points.

## Reporting summary

Further information on research design is available in the Nature Portfolio Reporting Summary linked to this article.

# Data availability

Source data of flow cytometry data are provided with this paper. The flow cytometry files (.fcs) generated for this project are available at the Figshare repository: https://doi.org/10.6084/m9.figshare.26661349.v1 and https://doi.org/10.6084/m9.figshare.26661358.v1. All raw imaging data and transgenic lines available on request. Source data are provided with this paper.

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

## Acknowledgements

We thank U. Technau and M. Gibson lab for sharing monoclonal and polyclonal NvVasa2 antibodies, Marie Montjouridès for experimental support contributing to the characterization of the *vasa2::mOr2* and *piwi1^mOr2* reporter lines, Fabian Rentzsch's lab at the University of Bergen for sharing *soxB(2)::mOr2, prdm14d::GFP, insm1::GFP* and *elav1::cerulean* reporter lines and for constructive feedback, and all Steinmetz lab

members for helpful discussions. The flow cytometry was performed at the Flow & Mass Cytometry Core Facility, Department of Clinical Science, University of Bergen. This work was funded by core budget (NFR grant 234817) of the Michael Sars Centre at the University of Bergen. E.P.-C. is funded by an EMBO fellowship (ALTF 406-2021).

## Author contributions

P.M.-P. designed the study, interpreted the results, performed ISH, EdU labelling and immunostainings, generated the *vasa2::mOr2* construct and the *piwi1^mOr2^* line, established and characterized all transgenic reporter lines, did scientific graphics and wrote the manuscript. E.P.-C. performed and analyzed all flow cytometry experiments, and contributed to writing the manuscript. P.R.H.S. designed the study, interpreted the results, wrote the manuscript and generated the CRISPR-Cas9 knock-in *piwi1^mOr2^* and *piwi1^P2A-GFP^* constructs and the *piwi1^P2A-GFP^* line.

## Funding

## Competing interests

The authors declare no competing interests.
