## [Peer Review File · Nature Communications]

A population of Vasa2 and Piwi1 expressing cells generates germ cells and neurons in a sea anemoneREVIEWER COMMENTS

Reviewer #1 (Remarks to the Author):

Germline segregation in early embryonic development is considered to be essential for protecting germ cells from mutations. This process is observed in most bilaterians, such as chordates, insects, and nematodes, but it has not been identified in plants, where somatic tissue can give rise to flowering meristems throughout life. The occurrence and evolutionary history of germline segregation in basal metazoans remain subjects of debate. -- In bilaterians, germline segregation takes place when primordial germ cells (PGCs) segregate from somatic cells during early embryogenesis, forming male or female germline stem cells (GSCs) that give rise to gametes in adult animals. However, in basal metazoans, a concept of adult stem cells has been proposed, retaining both somatic and germinal potentials throughout life, akin to plants. These 'primordial stem cells' (PriSCs) are primarily based on studies in cnidarians and planarians, particularly in hydra and related marine hydrozoans. Characterized by a distinctive drop-like shape, these cells, known as 'interstitial stem cells,' inhabit the interstitial space of both germ layers. While the existence of interstitial stem cells has been mainly explored in cnidarians, particularly in hydra, their identification in other cnidarians is yet to be confirmed and is currently under scrutiny. -- In the manuscript under review, the authors sought to identify PriSCs in *Nematostella vectensis*. Combining expression analyses of conserved germline and multipotency markers (e.g., Piwi, Vasa) with new Piwi1 and Vasa2 transgenic reporter lines, they demonstrated strong expression of these markers in germ line cells and weaker expression in neuronal progenitor cells. Although the identification of germ line cells and neuronal progenitor cells represents progress in recognizing stem cell populations in *Nematostella*, the assertion that Vasa2+/Piwi1+ cells constitute a population of juvenile and adult stem-like cells in *Nematostella*, generating gametes and a diversity of somatic cells, including neurons, is less convincing.

Major comments

Identity of Vasa2+/Piwi1+ populations.

The authors conducted a repetition of prior expression analyses (ISH) by the Gibson and Technau lab. These analyses demonstrated the expression of orthologous piwi1/2, vasa1/2, pl10, and tudor genes in germ line cells of adult male and female polyps. Utilizing EdU labelling in combination with Vasa2 antibodies and various transgenic strains, the authors identified proliferating germ line cells in mesenteries, delineating the gonadal location. Beyond gonadal localization, proliferating Vasa2+/Piwi1+ cells were also observed in non-gonadal cells of the basiepithelial region of the septal filaments.

For a more in-depth analysis of Vasa2+/Piwi1+ cells, the authors generated transgenic strains with a knock-in allele (piwi1mOr2) where the mOrange2 fluorophore was fused N-terminally to Piwi1. Additionally, an alternative vasa2::mOr2 reporter line was generated using I-SceI-meganuclease-mediated transgenesis. In all instances, Vasa2+/Piwi1+ cells exhibited robust reporter activity in the gonadal region and in developing gametes. Colocalization of Vasa2/mOr2 and Piwi1 was also observed in basiepithelial cells of nongonadal regions of the gastrodermis (pharynx, subpharynx, and trophic region) in both juvenile and adult polyps. However, these nongonadal cells exhibited significantly lower expression levels of Piwi1 and Vasa2 compared to gonadal cells.

The authors reported the presence of nongonadal cells containing mOr2-Piwi1 protein "near the detection limit" in different parts of juvenile and adult polyps (along the parietal muscle tract and in the body wall gastrodermis). The low protein levels of detectable Vasa2 or piwi1, or even their absence in these cells, are proposed to result from cytoplasmic inheritance from Vasa2+/Piwi1+ cells.

While this is a plausible scenario, alternative hypotheses should be considered. For instance, somatic stem cells and germline stem cells might represent independent stem cell lineages with varying levels of Vasa2/Piwi1 expression, such as high expression in germ line stem cells and low expression in somatic stem cells

Lineage tracing of Vasa2+/Piwi1+ cells.

To investigate potential cell lineages of extra-gonadal Vasa2+/Piwi1+ cells, the authors generated a piwi1(P2A-GFP) knock-in allele by introducing the P2A self-cleaving peptide between the GFP and the N-terminal region of Piwi1. This design allowed them to track the GFP+ progeny of Vasa2+/Piwi1+ cells in a Piwi1-independent manner. As a control they established a piwi1(P2A-GFP)/piwi1(mOr2) double reporter line. These are valuable tools and confirmed the co-localization of high levels of GFP and mOr2 reporters in developing germ line cells. -- Using these reporter lines to trace the origin of non-gonadal Vasa2+/Piwi1+ cells in adult and juvenile polyps, the authors observed a significant overlap between piwi1(P2A-GFP+) and vasa2::mOr2+ somatic cells. From this observation, they concluded that the progeny of Vasa2+/Piwi1+ cells does develop into both, gametes and a diversity of somatic cells throughout the polyp body column. Notably, based on the uniform levels of GFP throughout the epidermis of piwi1(P2A-GFP) animals, the authors even speculated that epidermis derives from mesenterial Vasa2+/Piwi1+ stem-like cells.

While these lineage tracing experiments represent an important piece of work, the interpretation of this data set strongly depends on the properties and homogeneity of the Vasa2+/Piwi1+ stem cell population. Is it homogeneous population or are there several ones, at least two, one germline and one somatic population, or rather several somatic stem cell populations?

The authors suggest that Vasa2+/Piwi1+ cells constitute a primordial stem cell population (PriSCs) with germline and somatic potentials, which is in contrast to previous views that Vasa2+/Piwi1+ cells are primordial germ cells (PGCs). However, without any further insight into the homo- or heterogeneity of the multipotent Vasa2+/Piwi1+ cell population, it cannot be ruled out that these cells are primordial stem cells, as previously hypothesized by Chen et al (2020) and Extavour et al (2005). The authors compare their results with previous findings in the freshwater polyp Hydra, where Vasa2+/Piwi1+ like stem cells have been described in the interstitial stem cell population (called i-cells). I-cells and Nematostella Vasa2+/Piwi1+ cells certainly share their basal epithelial location, expression of Vasa and Piwi, and formation of neuroglandular cell lineages including nematocytes, but there is an ongoing debate in Hydra as to whether the interstitial stem cells of Hydra can truly be considered a primordial stem cell population. Cell lineage studies and molecular work also point to the existence of a primordial germ cell population in Hydra. Thus, although there has been significant progress in the identification of stem cells in Nematostella, and similarities with the interstitial cells of hydrozoans may suggest common germline principles among cnidarians, it is premature and misleading to characterize the Vasa2+/Piwi1+ cell population as PriSCs.

Further comments

Quantitative data are missing.

One limitation of this study is the absence of quantitative data on distinct cell populations. Without establishing a quantitative relationship between various stem cell populations, understanding their function and dynamics becomes challenging, hindering the assessment of the reliability of proposed hypotheses. The authors mention cells unmarked even after a 3-day EdU pulse, categorizing them as quiescent or slow cyclers. Providing an estimate of these numbers would be valuable. The assertion that mesenteric stem cells exclusively sustain epithelia seems impractical given the global size ratios;

incorporating numerical data would lend support. Numbers are also crucial in the context of single-cell sequencing, where the proximity of a germline to a somatic stem cell line might easily be overlooked.

Text structure.

While the overall manuscript structure is clear and straightforward, the individual sections can be challenging to read due to the detailed inclusion of extensive data from the supplements. Additionally, there is room for improvement in the organization of the illustrations. For instance, in Figure 1, presenting RT in cross and side views side by side could enhance orientation. Furthermore, the ISH images are somewhat distracting in their current format and might be more effective if presented alongside the antibody data for a more coherent display.

Reviewer #2 (Remarks to the Author):

This is a well-presented study addressing the location and fate of cells expressing conserved germ line/multipotency genes, notably including Piwi and Vasa, in the anthozoan cnidarian *Nematostella*. This issue is of particular interest because despite the considerable regenerative properties of these animals, no adult stem cell populations have been identified - in marked contrast to the well-studied regenerative hydrozoan cnidarian *Hydra*.

The study elegantly uses a combination of careful in situ hybridizations, anti-Vasa immunostaining and various new transgenic reporter lines for *piwi1* and *vasa2*. As a proxy for lineage tracing, the authors analyse the distribution of fluorescent reporter proteins expressed from exogenous transgenes or generated by CRISPR-tagging of endogenous genes, detecting them persisting in a variety of somatic cell types as well as in germ cells.

The authors conclude from their observations that a "stem cell like population" is present in *Nematostella* adults and juveniles that generates germ cells as well as some somatic cell types including neural cells.

The second part of this conclusion - that cells expressing detectable Vasa and Piwi proteins positioned along the length of the internal mesenteries give rise to germ cells and also to somatic cell types - is an important finding and is convincingly demonstrated. The authors further made crosses with other transgenic lines to start to discriminate the different possible somatic fates of these cells. As they point out (lines 194-204), interpretation of these observations is not entirely clear cut. The main concern relates to detection threshold issues such that cells from unrelated lineages expressing Vasa or Piwi at low levels, close to or below detection levels, could artefactually accumulate the fluorescence proteins/tagged proteins to detectable levels. A particular example of this issue is in the juvenile epidermis and gastrodermis (Fig S5) where low constitutive expression certainly cannot be ruled out. Definitive proof that the strongly Vasa+/Piwi+ cells detected along the mesenteries generate particular somatic cell types will require cell tracking, eg by photoconversion of fluorescent markers in the presumed precursor population, or their ablation, in future studies. Overall, the authors present this issue fairly and the remaining areas of doubt do not detract from the value of this main conclusion.

More seriously, I question whether it is justified to define these Vasa+/Piwi+ mesentery cells as "stem-cell like". A defining property of stem cell populations is self-renewal, and this has not been demonstrated. Detection of EdU incorporation into Piwi+/Vasa+ mesentery cells immediately after EdU incubation is not proof of cell renewal. An alternative explanation for the data presented is that somatic cells positioned along the mesenteries (and possibly elsewhere) can embark on programmes of germ line or nerve cell differentiation which as an early step involve upregulation of the "conserved multi-potency gene set". Unless I am missing something here, such alternative explanations should be

discussed, caveats added regarding the “stem-cell like population” conclusion throughout the manuscript, and the title changed to something like “Piwi/Vasa-rich precursors generate germ cells and neurons in a sea anemone”.

Specific points :

1. I had difficulty integrating the information illustrated in Figures 1C-D, S1 and 2E-I concerning at what levels of the animal and at what stages mRNAs for Vasa and the other mRNAs were detected. I would appreciate if more comprehensive sets of in situ hybridization images for each gene in different regions could be added as a Supplementary Figure. A summary of which cells were positive for mRNA and protein for each gene would also be useful, perhaps as a table or short paragraph, explicitly pointing out any differences.
2. The labels on the tree in Fig. 1A are a mix of branch names vs selected examples of animal for some branches. It would be helpful for non-expert readers if the terminology could be made uniform. I would also encourage the authors to avoid the outdated term “phyla” in the legend, which can add to the confusion.
3. Scale bars are needed for Fig.1 C-D, Fig. S1 and Fig S6.
4. Fig S2 (B-B’; inlets of A-A’) “Weak staining in intermediate and distal ciliated tract cells using the monoclonal mouse anti-NvVasa2 might represent food vesicles and could not be reproduced.” Please explain in more detail what is meant by this. How many time was the experiment performed ? If it was more common to detect no staining in this region, why not show images from a more representative experiment?
Fig S2 (D-D’; inlets of C-C’) “Representative example of unspecific binding of both antibodies to putative mucus cells in the parietal muscle tract.” Please give the evidence that this is unspecific staining.
5. Lines 158 -161. Please state alternative possibilities, for instance that Piwi is expressed constitutively at low levels in self -renewing epidermal cells, as reported in Hydra.
6. The authors briefly cite the recent BioRxiv preprint by Denner et al (doi.org/10.1101/2023.12.07.570436) in their Introduction. It would be pertinent in the Discussion, to point out similarities and differences in the conclusions of the two studies (eg at lines 256-259).

Reviewer #3 (Remarks to the Author):

Miramón-Puértolas and Steinmetz describe a population of cells in the sea anemone *Nematostella* that express *piwi1*, *vasa1*, *vasa2*, *pl10*, and *tudor*. These genes belong to the so-called germline multipotency program (GMP; Juliano et al. 2010). In animals with embryonic germline segregation (e.g., flies, worms, and vertebrates), these genes are expressed in gamete progenitors. However, some invertebrates do not segregate a germline during embryogenesis but possess GMP+ adult stem cells that retain the potential to differentiate not only into gametes but also generate somatic cells. These cells have been termed ‘primordial stem cells’ (PriSCs) and their existence proposed to correlate with whole-body regenerative abilities (Solana 2013). Pluripotent or multipotent PriSCs have been confirmed in hydrozoan cnidarians (jellies and hydroids) but their existence in anthozoans (corals and sea anemones) remains uncertain. Because PriSCs are also found in planarians and acoels (confirmed only at a population level in the latter), the evolutionary history of these cells is unclear. They could either be a primitive character in animals, lost in lineages that segregate a germline, or evolved multiple times in groups without a segregated germline.

The authors performed in situ hybridization with probes targeting the *Nematostella* GMP genes *piwi1*, *vasa1*, *vasa2*, *pl10*, and *tudor*, and performed IF with anti-Vasa2 antibodies. They find cells (some in

S-phase) expressing these genes not only in the gonadal mesentery and maturing germ cells but also in basiepithelial cells along the reticulate tract of the mesentery; these cells are probably not germ cells due to their location relative to the gonad. The authors then used transgenic animals that express fluorescent proteins in a GMP context to track these cells and their derivatives that no longer express GMP genes, based on the carry-over of fluorescent proteins with a long half-life. The authors combined animals that contain knock-in fluorescent proteins in-frame with GMP genes, with and without a P2A peptide (i.e., fusion proteins or free FPs). They also generated a reporter animal that expresses GFP under the vasa2 promoter. The authors detect fluorescence in various somatic cells (they could only confirm neurons). Since these cells do not express GMP, their fluorescence must have been inherited from GMP+ progenitors.

Overall, the experiments are well performed and presented. The authors show that, at the population level, piwi1+/vasa2+ (i.e., GMP+) cells behave like multipotent PriSCs that can contribute to somatic lineages as well as to gamete production. Future studies should address the heterogeneity of these cells and their developmental potential at single-cell resolution, as done in planarians and hydrozoans. However, as a first step towards characterization anthozoan stem cells, this is an interesting report.

Minor comments are listed below.

Line 11 – add "and cnidarians" after "planarians". Cnidarians were studied well before planarians in this context.

Line 12 – replace "until" by "throughout".

Line 26 – "preformation or induction" is rather clunky (despite being historically correct). Consider replacing it by a more accessible term like "maternal or zygotic induction" (or something similar).

Line 46 – replace "phylum" by "class".

Line 80 – a short explanation of what mesenteries are would benefit the non-specialists.

Line 120 – remove the redundant "Vasa2+/Piwi1+".

Line 131 – not sure that "lineage tracing" is the correct term here because the heterogeneity of the population remains unknown. Multiple, lineage-restricted stem cells may constitute the Nematostella GMP+ pool.

Line 301 – "supports" refers to "data", which is plural; should be "support".

Line 305 – no need for quotation marks in "i-cells". This is an established term.

Line 342 – replace "piece of promoter" by "1.6 kb genomic fragment upstream the start codon...".

Line 345 – since it is unknown which part of the fragment constitutes the actual promoter/enhancer, use something like "the genomic fragment that included the regulatory elements...".

With respect to the authors' failure to verify contribution of GMP+ cells to nematogenesis, the authors may want to mention that Denner et al. (2023; <https://doi.org/10.1101/2023.12.07.570436>), who used a similar approach (tracking nanos2+ cells), do show contribution of GMP+ cells to nematocyte production (this preprint is cited but not in this context).

The authors report expression of GMP genes in “extra-gonadal location” (e.g., line 772). How are gonad boundaries defined in *Nematostella*?

Legend to Fig S1 – A-A” should be A-A'''

Fig 5J is redundant, already included in Fig 6.

With respect to the authors' outlook on cell-based approaches to engineer heat-resistant corals (lines 329-331), the authors may want to mention the preprints by Talice et al. (2022) <https://doi.org/10.21203/rs.3.rs-2137324/v1> who show the existence of putative stem cells in *Nematostella* and discuss it in this context.

Point-to-point response to reviewer comments

Note: Due to major rearrangements in the figures and to keep some clarity, we have used red labelling only to highlight major changes in the main text, but not in the legends of main or supplementary figures.

Responses to reviewer #1

Germline segregation in early embryonic development is considered to be essential for protecting germ cells from mutations. This process is observed in most bilaterians, such as chordates, insects, and nematodes, it has not been identified in plants, where somatic tissue can give rise to flowering meristems throughout life. The occurrence and evolutionary history of germline segregation in basal metazoans remain subjects of debate. — In bilaterians, germline segregation takes place when primordial germ cells (PGCs) segregate from somatic cells during early embryogenesis, forming male or female germline stem cells (GSCs) that give rise to gametes in adult animals. However, in basal metazoans, a concept of adult stem cells has been proposed, retaining both somatic and germinal potentials throughout life, akin to plants. These 'primordial stem cells' (PriSCs) are primarily based on studies in cnidarians and planarians, particularly in hydra and related marine hydrozoans. Characterized by a distinctive drop-like shape, these cells, known as 'interstitial stem cells,' inhabit the interstitial space of both germ layers. While the existence of interstitial stem cells has been mainly explored in cnidarians, particularly in hydra, their identification in other cnidarians is yet to be confirmed and is currently under scrutiny. — In the manuscript under review, the authors sought to identify PriSCs in *Nematostella vectensis*. Combining expression analyses of conserved germline and multipotency markers (e.g., Piwi, Vasa) with new Piwi1 and Vasa2 transgenic reporter lines, they demonstrated strong expression of these markers in germ line cells and weaker expression in neuronal progenitor cells.

1) We thank the reviewer for a thorough synopsis of our main results, which allowed us to understand that some of the reviewer's concerns are based on misunderstandings or misinterpretations of our mRNA and immunostaining results. In disagreement with the reviewer's synopsis, we found no Vasa2 protein or the mRNA of *vasa2*, *piwi1*, *GFP* or *mOr2* in any cells other than germ cells or *vasa2*⁺/*piwi1*⁺ cells in the reticulate tract (previously

termed 'stem-like'). The only protein we can still detect at lower levels in potential neuronal progenitor cells is Piwi1-mOr2, which in the absence of *piwi1* or *mOr2* mRNA is likely the result of cytoplasmic inheritance as we discussed.

Nevertheless, some of the comments and concerns from reviewer #1 and #2 clearly indicated a necessity to increase the clarity and readability of our manuscript. We have therefore enhanced and streamlined the graphical overviews summarizing gene and protein expression of transgenic lines (see also response 10). We redesigned Fig. 6 and added an overall graphical summary of the expression of all proteins, mRNAs and transgenes in the germline or somatic cells (Fig. 6A). In addition, we added an Excel table as Supplementary Data 1 to summarize the location and relative, approximate expression levels of all mRNAs and proteins studied in this manuscript.

Although the identification of germ line cells and neuronal progenitor cells represents progress in recognizing stem cell populations in *Nematostella*, the assertion that Vasa2+/Piwi1+ cells constitute a population of juvenile and adult stem-like cells in *Nematostella*, generating gametes and a diversity of somatic cells, including neurons, is less convincing.

Major comments

Identity of Vasa2+/Piwi1+ populations.

The authors conducted a repetition of prior expression analyses (ISH) by the Gibson and Technau lab. These analyses demonstrated the expression of orthologous *piwi1/2*, *vasa1/2*, *pl10*, and *tudor* genes in germ line cells of adult male and female polyps. Utilizing EdU labelling in combination with Vasa2 antibodies and various transgenic strains, the authors identified proliferating germ line cells in mesenteries, delineating the gonadal location. Beyond gonadal localization, proliferating Vasa2+/Piwi1+ cells were also observed in non-gonadal cells of the basiepithelial region of the septal filaments.

For a more in-depth analysis of Vasa2+/Piwi1+ cells, the authors generated transgenic strains with a knock-in allele (*piwi1mOr2*) where the mOrange2 fluorophore was fused N-terminally to Piwi1. Additionally, an alternative *vasa2::mOr2* reporter line was generated using I-SceI-meganuclease-mediated transgenesis. In all instances, Vasa2+/Piwi1+ cells exhibited robust reporter activity in the gonadal region and in developing gametes.

Colocalization of Vasa2/mOr2 and Piwi1 was also observed in basiepithelial cells of nongonadal regions of the gastrodermis (pharynx, subpharynx, and trophic region) in both juvenile and adult polyps. However, these nongonadal cells exhibited significantly lower expression levels of Piwi1 and Vasa2 compared to gonadal cells.

2) We kindly note that the reviewer's synopsis disagrees with our results: we did not show that non-gonadal cells exhibit lower expression levels of Piwi1 and Vasa2 than gonadal cells. As can be seen on Fig. 1I, I', Fig. 3A or Supplementary Fig. 2, the signal of both mOr2-Piwi1 or Vasa2 proteins in spermaries (Fig. 1I, I') or oocytes (Fig. 3A, Supplementary Fig. 2) are on one image within a similar dynamic range than found in non-gonadal cells.

In order to avoid misunderstandings in relative expression levels, we now consistently indicate relative expression levels of fluorescent signals in both in the text and in figures (e.g. [mOr2-Piwi1]_{low} or [mOr2-Piwi1]_{high}). Along those lines, we realised that the term 'Piwi1+ cells' is misleading as cells can have high or low Piwi1-mOr2 levels. In order to be more precise, we therefore consistently changed 'Vasa2+/Piwi1+' to 'Vasa2+[mOr2-Piwi1]_{high}' throughout the results part and in all figures. We kept it in the discussion (and clearly explain it, l. 292-3) for simplicity to refer to a specific cell population of cells in the reticulate tract.

The authors reported the presence of nongonadal cells containing mOr2-Piwi1 protein "near the detection limit" in different parts of juvenile and adult polyps (along the parietal muscle tract and in the body wall gastrodermis). The low protein levels of detectable Vasa2 or piwi1, or even their absence in these cells, are proposed to result from cytoplasmic inheritance from Vasa2+/Piwi1+ cells. While this is a plausible scenario, alternative hypotheses should be considered. For instance, somatic stem cells and germline stem cells might represent independent stem cell lineages with varying levels of Vasa2/Piwi1 expression, such as high expression in germ line stem cells and low expression in somatic stem cells

3) We agree that the mOr2 or GFP protein detected in parietal muscle tract, body wall gastrodermis or epidermis could result from independent *vasa2*- and *piwi1*-expressing cell populations and believe that we had addressed this point when discussing the expression of fluorophore in the epidermis. We also kindly note that contrary to the

reviewer's comment, we find no trace of Vasa2 protein or mRNA for *vasa2*, *piwi1*, *mOr2* or *GFP* in any of these regions in transgenic lines or wildtypes. In order to avoid any misunderstandings, we have refined and expanded on this notion and alternative scenarios are now presented at several points in the results (l. 140-148; l. 191-194; 242-245; 280-283) and discussion (l. 369-273).

Lineage tracing of Vasa2+/Piwi1+ cells.

To investigate potential cell lineages of extra-gonadal Vasa2+/Piwi1+ cells, the authors generated a *piwi1*(P2A-GFP) knock-in allele by introducing the P2A self-cleaving peptide between the GFP and the N-terminal region of Piwi1. This design allowed them to track the GFP+ progeny of Vasa2+/Piwi1+ cells in a Piwi1-independent manner. As a control they established a *piwi1*(P2A-GFP)/*piwi1*(mOr2) double reporter line. These are valuable tools and confirmed the co-localization of high levels of GFP and mOr2 reporters in developing germ line cells.

4) We would kindly add to the reviewer's synopsis that importantly, we also reported high levels of GFP and mOr2 in non-germline cells of adults and juveniles, and in regions that will never harbour germ cells (pharynx, trophic region).

Using these reporter lines to trace the origin of non-gonadal Vasa2+/Piwi1+ cells in adult and juvenile polyps, the authors observed a significant overlap between *piwi1*(P2A-GFP+) and *vasa2::mOr2*+ somatic cells. From this observation, they concluded that the progeny of Vasa2+/Piwi1+ cells does develop into both, gametes and a diversity of somatic cells throughout the polyp body column. Notably, based on the uniform levels of GFP throughout the epidermis of *piwi1*(P2A-GFP) animals, the authors even speculated that epidermis derives from mesenterial Vasa2+/Piwi1+ stem-like cells.

While these lineage tracing experiments represent an important piece of work, the interpretation of this data set strongly depends on the properties and homogeneity of the Vasa2+/Piwi1+ stem cell population. Is it homogeneous population or are there several ones, at least two, one germline and one somatic population, or rather several somatic stem cell populations?

5) We thank the reviewer for raising this important concern and, as we had clearly stated in the discussion, we fully agree with the reviewer that we cannot distinguish if the

population of Piwi1+/Vasa2+ cells is homogenous or heterogenous. We acknowledge however that there was room for confusion or misunderstanding in other parts of the manuscript. Throughout the manuscript, we therefore changed formulations so that the term 'multipotent' only refers to a 'population' of cells. In addition, we largely expanded the paragraph and alongside added a schematic (Fig. 6B) to discuss the stemness of cells, their potency and the homo-/heterogeneity of the Piwi1+/Vasa2+ population (l. 369-273).

The authors suggest that Vasa2+/Piwi1+ cells constitute a primordial stem cell population (PriSCs) with germline and somatic potentials, which is in contrast to previous views that Vasa2+/Piwi1+ cells are primordial germ cells (PGCs). However, without any further insight into the homo- or heterogeneity of the multipotent Vasa2+/Piwi1+ cell population, it cannot be ruled out that these cells are primordial stem cells, as previously hypothesized by Chen et al (2020) and Extavour et al (2005).

6) We thank the reviewer for raising the concern and for giving us the opportunity to clarify this point. We kindly note that the previously published assumption that Vasa2+/Piwi1+ cells are primordial germ cells (see papers cited by the reviewer) is based on ISH and immunostaining on fixed animals, which are generally very limited to provide any information about cell lineages. We therefore believe that our experiments, with all their limitations, are more informative than previous reports and show that it is very unlikely that the Vasa2+/Piwi1+ cell population contributes only to non-germinal cells, as expected if they consist of PGCs. We agree however that we cannot rule out that a subset of Vasa2+/Piwi1+ cells constitutes a pool of unipotent PGCs, but then another subset must be at the origin of somatic cells. We include this possibility as part of the discussion on the homo-/heterogeneity of the Vasa2+/Piwi1+ cell population in (l. 369-273) and Fig. 6C.

The authors compare their results with previous findings in the freshwater polyp Hydra, where Vasa2+/Piwi1+ like stem cells have been described in the interstitial stem cell population (called i-cells). I-cells and *Nematostella* Vasa2+/Piwi1+ cells certainly share their basal epithelial location, expression of Vasa and Piwi, and formation of neuroglandular cell lineages including nematocytes, but there is an ongoing debate in Hydra as to whether the interstitial stem cells of Hydra can truly be considered a primordial stem cell population. Cell lineage studies and molecular work also point to the

existence of a primordial germ cell population in Hydra. Thus, although there has been significant progress in the identification of stem cells in *Nematostella*, and similarities with the interstitial cells of hydrozoans may suggest common germline principles among cnidarians, it is premature and misleading to characterize the Vasa2+/Piwi1+ cell population as PriSCs.

7) We thank the reviewer for raising this concern and acknowledge that it was premature to name this population of cells as 'PriSCs'. We therefore now refrain from naming Piwi1+/Vasa2+ as 'stem-like' cells throughout the manuscript. In the discussion, we have kept the comparison with i-cells, which we think is still informative. We have removed the notion that *Nematostella vasa2+/piwi1+* are PriSCs, but added to the outlook the possibility that these cells are 'candidates for adult stem cells in a sea anemone.' (l. 393).

Further comments

Quantitative data are missing.

One limitation of this study is the absence of quantitative data on distinct cell populations. Without establishing a quantitative relationship between various stem cell populations, understanding their function and dynamics becomes challenging, hindering the assessment of the reliability of proposed hypotheses. The authors mention cells unmarked even after a 3-day EdU pulse, categorizing them as quiescent or slow cyclers. Providing an estimate of these numbers would be valuable. The assertion that mesenteric stem cells exclusively sustain epithelia seems impractical given the global size ratios; incorporating numerical data would lend support. Numbers are also crucial in the context of single-cell sequencing, where the proximity of a germline to a somatic stem cell line might easily be overlooked.

8) We thank the reviewer for this suggestion. We used flow cytometry on dissociated Piwi1^{mOr2} juveniles and adults to quantify the proportion of Vasa2+/Piwi1+ cells and their putative progeny. In addition, we quantified the EdU+ index within those cell populations. We have added this data as Fig. 2K-N, the underlying flow cytometry gating strategy as Supplementary Figure 5 and the numerical data as Excel sheet in Supplementary Data 2. The quantification confirmed the scarcity of the Vasa2+/Piwi1+ cells in juveniles and adults and the abundance of potential progeny cells. We therefore also included the 'rare'

aspect of these cells in the title and abstract. We note however that, as mentioned in the manuscript, our transgenic lines likely do not label complete cells lineages. They are therefore limited in estimating the numerical ratio of potential stem/progenitor and progeny, as requested by the reviewer.

Text structure.

While the overall manuscript structure is clear and straightforward, the individual sections can be challenging to read due to the detailed inclusion of extensive data from the supplements.

9) Based on the reviewer's comment, we have now removed some of the details when citing the supplementary figures, broadly cite whole figures and only kept citing subparts of a supplementary figure if we deemed it absolutely necessary.

Additionally, there is room for improvement in the organization of the illustrations. For instance, in Figure 1, presenting RT in cross and side views side by side could enhance orientation. Furthermore, the ISH images are somewhat distracting in their current format and might be more effective if presented alongside the antibody data for a more coherent display.

10) We thank the reviewer for this helpful comment, which motivated us to improve the organisation, coherence and clarity of the figures and schematics. Major changes are:

- Fig.1 was reorganised to present ISH alongside antibody stainings as suggested by the reviewer.
- We moved all Vasa2/mOr2-Piwi1 data on adults from Fig. 2 into Fig1, resulting in a coherent separation between adult (Fig. 1) and juvenile (Fig. 2) datasets.
- In Fig. 3,D, F, H, we increased the brightness to improve visibility of low mOr2 levels
- In Fig.4., the mesentery and spermary overviews (previously Fig.4A, B) were redundant with the rest of the figure and removed.
- The number of Supplementary Figures was reduced by integrating the nuage/perinuclear granule data from Supplementary Fig. 4 (now defunct) into S3J-K or as inlets into Fig. 1F and S1E.

- We removed technical data (transgenesis or knock-in allele schematics) from the main figures into a new Supplementary Fig. 4.

- We redesigned the schematics in the main and supplementary figures using a clear colour code and avoiding redundancies between schematics (as also noted by reviewer #2). We also added the notions of 'high' and 'low' expression of transgenes for the three main groups of cells (putative 'stem/progenitor cells', germline, putative 'somatic progeny') to all relevant figures.

- We redesigned Fig. 6 to provide a comprehensive summary of gene & protein expression levels, and of the different models of cell lineages.

Reviewer #2

This is a well-presented study addressing the location and fate of cells expressing conserved germ line/multipotency genes, notably including Piwi and Vasa, in the anthozoan cnidarian *Nematostella*. This issue is of particular interest because despite the considerable regenerative properties of these animals, no adult stem cell populations have been identified - in marked contrast to the well-studied regenerative hydrozoan cnidarian *Hydra*.

The study elegantly uses a combination of careful in situ hybridizations, anti-Vasa immunostaining and various new transgenic reporter lines for *piwi1* and *vasa2*. As a proxy for lineage tracing, the authors analyse the distribution of fluorescent reporter proteins expressed from exogenous transgenes or generated by CRISPR-tagging of endogenous genes, detecting them persisting in a variety of somatic cell types as well as in germ cells.

The authors conclude from their observations that a "stem cell like population" is present in *Nematostella* adults and juveniles that generates germ cells as well as some somatic cell types including neural cells.

The second part of this conclusion - that cells expressing detectable Vasa and Piwi proteins positioned along the length of the internal mesenteries give rise to germ cells and also to somatic cell types - is an important finding and is convincingly demonstrated. The authors further made crosses with other transgenic lines to start to discriminate the different possible somatic fates of these cells. As they point out (lines 194-204),

interpretation of these observations is not entirely clear cut. The main concern relates to detection threshold issues such that cells from unrelated lineages expressing Vasa or Piwi at low levels, close to or below detection levels, could artefactually accumulate the fluorescence proteins/tagged proteins to detectable levels. A particular example of this issue is in the juvenile epidermis and gastrodermis (Fig S5) where low constitutive expression certainly cannot be ruled out. Definitive proof that the strongly Vasa+/Piwi+ cells detected along the mesenteries generate particular somatic cell types will require cell tracking, eg by photoconversion of fluorescent markers in the presumed precursor population, or their ablation, in future studies. Overall, the authors present this issue fairly and the remaining areas of doubt do not detract from the value of this main conclusion.

More seriously, I question whether it is justified to define these Vasa+/Piwi+ mesentery cells as “stem-cell like”. A defining property of stem cell populations is self-renewal, and this has not been demonstrated. Detection of EdU incorporation into Piwi+/Vasa+ mesentery cells immediately after EdU incubation is not proof of cell renewal. An alternative explanation for the data presented is that somatic cells positioned along the mesenteries (and possibly elsewhere) can embark on programmes of germ line or nerve cell differentiation which as an early step involve upregulation of the “conserved multipotency gene set”. Unless I am missing something here, such alternative explanations should be discussed, caveats added regarding the “stem-cell like population” conclusion throughout the manuscript, and the title changed to something like “Piwi/Vasa-rich precursors generate germ cells and neurons in a sea anemone”.

11) We kindly refer the reviewer to our responses 1), 5), 6), 7) and 10) to reviewer #1 that address these concerns.

We have also removed the 'stem-like' notion from the manuscript (incl. title) and changed it to 'A rare, multipotent population of Vasa2+/Piwi1+ cells generates germline and neurons in a sea anemone'.

Specific points :

1. I had difficulty integrating the information illustrated in Figures 1C-D, S1 and 2E-I concerning at what levels of the animal and at what stages mRNAs for Vasa and the other mRNAs were detected. I would appreciate if more comprehensive sets of in situ

hybridization images for each gene in different regions could be added as a Supplementary Figure. A summary of which cells were positive for mRNA and protein for each gene would also be useful, perhaps as a table or short paragraph, explicitly pointing out any differences.

12) We thank the reviewer for this comment that helped us to communicate this complex dataset in a better and clearer way. We kindly refer the reviewer to response 10) for an overview of major changes in figures aimed at a clearer separation of adult and juvenile data. In addition, as proposed by the reviewer, we have included a tables as new Supplementary Data 1 that provides a comprehensive overview of the location, relative levels of all mRNA and proteins studied in the manuscript. This also clarifies regions for which we have no expression signal or missing data.

We have decided to refrain from adding a large set of ISH images from additional parts of the animals for the following reasons:

- due to the potential of post-transcriptional and post-translational control, we strongly felt that an in-depth analysis of the localisation of Vasa2 and Piwi1 proteins, as f.ex. present for the entire mesentery (Supplementary Fig. 3), tentacles or the epidermis (Supplementary Fig. 8), is more informative than of their mRNA.

- As the paper is already data-rich, adding a larger set of additional ISH images risks to be counter-productive in our efforts to make the manuscript better readable and accessible for a non-expert reader.

- Technically, we kindly ask the reviewer to take into consideration that performing ISH on adult tissues is still a very challenging and time-consuming task in *Nematostella*. Due to the size of adults (1-5 cm long) and juveniles (5-15mm), ISH on those stages is currently only possible after dissection of pieces of interest. Performing ISH of all genes and all subparts of adult and juvenile polyp would certainly take many more months if not years, likely without a major gain of information.

All-in-all, we therefore hope that the reviewer understands our decision to prioritize the in-depth analysis of the protein and transgene location rather than of the mRNA. Ultimately, the method of choice to get a fully comprehensive view on the expression of GMP markers throughout all cell types and tissues will be a single-cell RNA sequencing

analysis with a much deeper sequence depths and higher cell numbers as currently available in the *Nematostella* literature.

2. The labels on the tree in Fig. 1A are a mix of branch names vs selected examples of animal for some branches. It would be helpful for non-expert readers if the terminology could be made uniform. I would also encourage the authors to avoid the outdated term “phyla” in the legend, which can add to the confusion.

13) We thank the reviewer for raising these concerns. We changed both the branch names and replaced the term 'phyla' by 'taxa' throughout the manuscript.

3. Scale bars are needed for Fig.1 C-D, Fig. S1 and Fig S6.

14) Scale bars have been added in all the figures mentioned (now Fig. 1E-F, S1 and S6).

4. Fig S2 (B-B”; inlets of A-A”) “Weak staining in intermediate and distal ciliated tract cells using the monoclonal mouse anti-NvVasa2 might represent food vesicles and could not be reproduced.” Please explain in more detail what is meant by this. How many time was the experiment performed ? If it was more common to detect no staining in this region, why not show images from a more representative experiment?

15) We apologize for highlighting a non-reproducible aspect of the Vasa2 antibody staining in this figure. NvVasa2 staining in the ciliated and/or intermediate tract can be seen without 'weak background' stain on multiple occasions throughout the manuscript (e.g. Fig. 2F, S3G", S6D, S9G). As the pictures represented in Supplementary Fig. 2 were best in sectioning quality and most representative for the purpose of showing the complete overlap of the two independently generated antibodies, we decided to keep the images but remove the potentially confusing reference to the low signal in these regions. Also, we believe that the weak background reinforces our point that even at the limits of the signal/noise ratio, no Vasa2 protein could be detected with any of the two antibodies within any of the regions where we detected weak fluorescent signal in the transgenes (e.g. body wall gastrodermis, epidermis, parietal muscle regions).

Fig S2 (D-D”; inlets of C-C”) “Representative example of unspecific binding of both antibodies to putative mucus cells in the parietal muscle tract.” Please give the evidence that this is unspecific staining.

16) We thank the reviewer for addressing this concern, which led us to clarify in the figure legend of Supplementary Fig. 2 that based on their morphology and location, these cells are most likely mucus cells. Most importantly, we noted that none of the cells expressing any of our transgenes in this region share a similar morphology.

5. Lines 158 -161. Please state alternative possibilities, for instance that Piwi is expressed constitutively at low levels in self-renewing epidermal cells, as reported in Hydra.

17) We kindly refer the reviewer to our response 3). Shortly, although we had discussed this possibility in our previous discussion, we have added this alternative scenario to the part pointed out by the reviewer (l. 191-194).

6. The authors briefly cite the recent BioRxiv preprint by Denner et al (doi.org/10.1101/2023.12.07.570436) in their Introduction. It would be pertinent in the Discussion, to point out similarities and differences in the conclusions of the two studies (eg at lines 256-259).

18) We thank the reviewer for this suggestion but decided to include only some of the conclusions from this preprint into our discussion. We are especially sceptical about any conclusions resulting from the *nanos2*-driven transgenic line. The reason is that the authors do not show any data beyond the gonad on the location of *nanos2* mRNA or protein in juveniles and adults. Without this data, however, it is not possible to distinguish the *nanos2::mOr2+* cells with endogenous *nanos2* expression from their progeny that carry *mOr2+* without expressing the *nanos2* gene. It remains also unclear if our *Vasa2+/Piwi1+* cells are expressing *nanos2* or labelled by the *nanos2::mOr2* transgenic line, which could have been solved by a *Vasa2* co-immunolabeling. This would have been important, especially as our unpublished single-cell data indicates that *nanos2*, in contrast to *piwi1* and *vasa2*, is not specifically enriched in GMP+ cell clusters, but rather broadly expressed in a wide range of progenitor cells. As we believe that it is not our role to discuss or review the flaws of this preprint, we hope that the reviewers understand that we decided to refrain from discussing the authors' *nanos2* data.

The *piwi1* transgenic line, instead, has provided some interesting additional information. Generated using I-SceI meganuclease, it likely labels a more complete cell lineage of *piwi1+* cells due to the likely integration of concatemers and results in higher levels of

transgene expression, as for our *vasa2::mOr2* line. Interestingly, the authors detect *piwi1::mOr2* protein in cnidocytes and gland cells, which are not labelled in our knock-in lines. Referring to this data, we have therefore added the notion that cnidocytes and gland cells might derive from *piwi1+/*vasa2+** cells in our manuscript (l. 338-347).

Reviewer #3

Miramón-Puértolas and Steinmetz describe a population of cells in the sea anemone *Nematostella* that express *piwi1*, *vasa1*, *vasa2*, *pl10*, and *tudor*. These genes belong to the so-called germline multipotency program (GMP; Juliano et al. 2010). In animals with embryonic germline segregation (e.g., flies, worms, and vertebrates), these genes are expressed in gamete progenitors. However, some invertebrates do not segregate a germline during embryogenesis but possess GMP+ adult stem cells that retain the potential to differentiate not only into gametes but also generate somatic cells. These cells have been termed 'primordial stem cells' (PriSCs) and their existence proposed to correlate with whole-body regenerative abilities (Solana 2013). Pluripotent or multipotent PriSCs have been confirmed in hydrozoan cnidarians (jellies and hydroids) but their existence in anthozoans (corals and sea anemones) remains uncertain. Because PriSCs are also found in planarians and acoels (confirmed only at a population level in the latter), the evolutionary history of these cells is unclear. They could either be a primitive character in animals, lost in lineages that segregate a germline, or evolved multiple times in groups without a segregated germline.

The authors performed in situ hybridization with probes targeting the *Nematostella* GMP genes *piwi1*, *vasa1*, *vasa2*, *pl10*, and *tudor*, and performed IF with anti-Vasa2 antibodies. They find cells (some in S-phase) expressing these genes not only in the gonadal mesentery and maturing germ cells but also in basiepithelial cells along the reticulate tract of the mesentery; these cells are probably not germ cells due to their location relative to the gonad. The authors then used transgenic animals that express fluorescent proteins in a GMP context to track these cells and their derivatives that no longer express GMP genes, based on the carry-over of fluorescent proteins with a long half-life. The authors combined animals that contain knock-in fluorescent proteins in-frame with GMP genes, with and without a P2A peptide (i.e., fusion proteins or free FPs). They also

generated a reporter animal that expresses GFP under the vasa2 promoter. The authors detect fluorescence in various somatic cells (they could only confirm neurons). Since these cells do not express GMP, their fluorescence must have been inherited from GMP+ progenitors.

Overall, the experiments are well performed and presented. The authors show that, at the population level, piwi1+/vasa2+ (i.e., GMP+) cells behave like multipotent PriSCs that can contribute to somatic lineages as well as to gamete production. Future studies should address the heterogeneity of these cells and their developmental potential at single-cell resolution, as done in planarians and hydrozoans. However, as a first step towards characterization anthozoan stem cells, this is an interesting report.

Minor comments are listed below.

Line 11 – add "and cnidarians" after "planarians". Cnidarians were studied well before planarians in this context.

Done.

Line 12 – replace "until" by "throughout".

Done.

Line 26 – "preformation or induction" is rather clunky (despite being historically correct). Consider replacing it by a more accessible term like "maternal or zygotic induction" (or something similar).

We agree and, as suggested by the reviewer, we replaced 'preformation or induction' by 'maternal or zygotic induction'.

Line 46 – replace "phylum" by "class".

We have replaced phylum/phyla by taxon/taxa here and throughout the manuscript.

Line 80 – a short explanation of what mesenteries are would benefit the non-specialists.

We thank the reviewer for this suggestion, and we have added ', which are folds of inner epithelium that harbour the developing gametes.

Line 120 – remove the redundant "Vasa2+/Piwi1+".

The part has been reformulated to include the new quantitative data.

Line 131 – not sure that "lineage tracing" is the correct term here because the heterogeneity of the population remains unknown. Multiple, lineage-restricted stem cells may constitute the *Nematostella* GMP+ pool.

As discussed in previous comments (see responses 5) and 6), we are aware of the potential heterogeneity of the 'GMP+ pool' and have clarified this in changes to the manuscript and figures. In that sense, we do not claim to perform lineage tracing on a single homogenous cell population, but to trace the lineages of a potentially heterogenous population of cells. For that reason, we always refer to a 'population' of cells. In the absence of a more precise term, we decided to continue using 'lineage tracing' and hope that it became clear enough that we do not imply that the stem/progenitor cell population is homogenous.

Line 301 – "supports" refers to "data", which is plural; should be "support".

Fixed.

Line 305 – no need for quotation marks in "i-cells". This is an established term.

Changed.

Line 342 – replace “piece of promoter” by “1.6 kb genomic fragment upstream the start codon...”.

Changed to 'A 1.6kb-long genomic fragment upstream of the putative *vasa2* transcription start site'

Line 345 – since it is unknown which part of the fragment constitutes the actual promoter/enhancer, use something like "the genomic fragment that included the regulatory elements...".

Changed as suggested.

With respect to the authors' failure to verify contribution of GMP+ cells to nematogenesis, the authors may want to mention that Denner et al. (2023; <https://doi.org/10.1101/2023.12.07.570436>), who used a similar approach (tracking

nanos2+ cells), do show contribution of GMP+ cells to nematocyte production (this preprint is cited but not in this context).

We kindly refer to our response #18 (Reviewer #2).

The authors report expression of GMP genes in “extra-gonadal location” (e.g., line 772). How are gonad boundaries defined in Nematostella?

Along the lines of previous publications, we have defined the gonad as the part of the mesentery where oocytes and spermatogonia are located. The precise borders of the gonad – the retractor muscle proximally and the reticulate tract distally – have been indicated in the schematics in Fig. 1D.

Legend to Fig S1 – A-A’ should be A-A’’

Corrected.

Fig 5J is redundant, already included in Fig 6.

We kindly refer to the response 10) for details about the changes we did to the schematics to remove redundancies and increase clarity.

With respect to the authors' outlook on cell-based approaches to engineer heat-resistant corals (lines 329-331), the authors may want to mention the preprints by Talice et al. (2022) <https://doi.org/10.21203/rs.3.rs-2137324/v1> who show the existence of putative stem cells in Nematostella and discuss it in this context.

We thank the reviewer for pointing us to this preprint that we have now cited accordingly (l. 399).

REVIEWERS' COMMENTS

Reviewer #1 (Remarks to the Author):

The authors have thoroughly revised the manuscript, taking into account the comments, criticisms and suggestions of the reviewers and incorporating various editorial suggestions. The current version is very readable and shows that Vasa2+/Piwi1+ cells are expressed in the germline and in somatic progenitors of neurons. These are exciting new data as they suggest that stem cells in *Nematostella* and *Hydra*, the two most intensively studied molecular genetic models of Cnidaria, are molecularly much more similar at the stem cell level than previously thought. This will bring together data sets from different cnidarian clades and may provide the much-needed common picture of how germline and somatic stem cells are molecularly controlled at the base of metazoan evolution. However, it is not yet possible to make a conclusive statement based on the current data (see discussion and Figure 6), more cell lineage analyses are lacking, including data on putative germ line segregation in embryos and adults.

For me, there are two points in this manuscript that should be considered again before publication, these are the title and the layout of Figure 6.

Title. The authors have changed the title based on comments from reviewer 2 and myself. On the one hand, this was in response to reviewer 2's critical question as to whether the Vasa+/Piwi+ cells can be described as "stem cell-like", as no experiment shows self-renewal of the cells. Second, this was due to my suggestion to include quantitative data for the different cell populations. Following the suggestion to quantify the cell populations, the authors performed flow cytometry on dissociated Piwi1mOr2 juveniles and adults to determine the proportion of Vasa2+/Piwi1+ cells and their putative progeny, and also determined the labelling index for S-phase cells (EdU+/EdU-) within these cell populations (Figure 2K-N, Figure S5, Table S2). These data show that Vasa2+/Piwi1+ cells are rare in juvenile and adult animals. Although these are important new data, I would not include the fact that they are "rare" in the title. Since the transgenic lines may not mark all cell lines, I think it is appropriate to make this point clear in the manuscript and discussion. To include "rare" in the title more quantitative data or cell line studies would be necessary to provide a robust answer. Instead, I recommend modifying Figure 6 and toning down the statements here.

Figure 6: This figure is overloaded, shows too many details and is redundant. The redundancy is within 6B but also between 6A and 6B. Useful is 6A with the summary of gene expression. 6B is redundant to 6A in terms of cellular differentiation trees, stem cell types (multipotent, unipotent) and experimentally demonstrated and speculative somatic differentiation events (plus: gamete/nerve vs question mark: nematocyte/gland cell/epidermis). One could dispense with 6B altogether if the three variants of multipotent cell populations were only described in the text, which is very clear here. The comments 'poorly-supported' and 'well-supported' for the "primordial germ cell model" and multipotent cell population model" are also part of the discussion and are less helpful for a summary figure, as they are highly dependent on interpretation.

Further comments

Some figure legends are overloaded with text, as they are often linked to statements similar to the main text. Ideally, the statement should be condensed in the figure captions, and the text should only contain descriptions for understanding the experiment.

In view of the importance of the reticular tract, a brief description of this tissue and its morphogenetic potential should be given at an appropriate place.

Reviewer #2 (Remarks to the Author):

This manuscript has been considerably clarified and improved in response to the reviewers' comments.

I appreciate the authors' efforts, including removing "stem cell like" from the conclusions and title, and the explicit framing of alternative hypotheses to explain their observations. The new summary diagram (Figure 6) and the overview in a new supplementary table of what observations have (and have not) been done are also helpful.

Specific minor points:

1. There are places where rewording of the added text is needed. As the authors mostly are careful to point out, lack of detection of proteins or mRNA does not necessarily equate to their absence, but depends on thresholds of detection. Eg Line 26 and line 141 should be changed eg to "absence of detectable" and "devoid of detectable")
2. The authors now include in the manuscript quantification of cells with high versus low levels of Piwi and Vasa proteins. This is done by flow cytometry of dissociated cells from [mOr2-Piwi1+] animals. How do the authors know that the [mOr2-Piwi1+] high vs [mOr2-Piwi1+] low cell populations detected this way can be equated to those that they defined by fluorescence microscopy methods? Some justification should be provided, or the assumptions stated explicitly.
3. Word missing on line 374 ('...data from stony corals...')

Reviewer #3 (Remarks to the Author):

The authors have adequately addressed the points I had raised in the previous review cycle. The manuscript is now very much improved. One minor suggestion for the usage of the term 'germline' in the new title and elsewhere: the term 'germline' is usually used to refer to segregated germ cell lineages (as seen in ecdysozoans and vertebrates). Here, the authors provide evidence that *Nematostella* possesses adult stem cells that can give rise to gametes and to at least some somatic cells, i.e., its germ cells are not segregated from somatic cells. Therefore, 'gametes' or 'germ cells', but not 'germline', seem more appropriate when discussing non-segregated germ cells. Overall, this manuscript the first major step in characterizing anthozoan stem cells and will have a profound impact on future studies on stem cells and their evolution.

Manuscript NCOMMS-24-05128: A population of Vasa2 and Piwi1 expressing cells generates germ cells and neurons in a sea anemone.

Point-to-point response to reviewer comments (final revisions)

Responses to reviewer #1

The authors have thoroughly revised the manuscript, taking into account the comments, criticisms and suggestions of the reviewers and incorporating various editorial suggestions. The current version is very readable and shows that Vasa2+/Piwi1+ cells are expressed in the germline and in somatic progenitors of neurons. These are exciting new data as they suggest that stem cells in *Nematostella* and *Hydra*, the two most intensively studied molecular genetic models of Cnidaria, are molecularly much more similar at the stem cell level than previously thought. This will bring together data sets from different cnidarian clades and may provide the much-needed common picture of how germline and somatic stem cells are molecularly controlled at the base of metazoan evolution. However, it is not yet possible to make a conclusive statement based on the current data (see discussion and Figure 6), more cell lineage analyses are lacking, including data on putative germ line segregation in embryos and adults.

For me, there are two points in this manuscript that should be considered again before publication, these are the title and the layout of Figure 6.

Title. The authors have changed the title based on comments from reviewer 2 and myself. On the one hand, this was in response to reviewer 2's critical question as to whether the Vasa+/Piwi+ cells can be described as "stem cell-like", as no experiment shows self-renewal of the cells. Second, this was due to my suggestion to include quantitative data for the different cell populations. Following the suggestion to quantify the cell populations, the authors performed flow cytometry on dissociated Piwi1mOr2 juveniles and adults to determine the proportion of Vasa2+/Piwi1+ cells and their putative progeny, and also determined the labelling index for S-phase cells (EdU+/EdU-) within these cell populations (Figure 2K-N, Figure S5, Table S2). These data show that Vasa2+/Piwi1+ cells are rare in juvenile and adult animals. Although these are important new data, I would not include the fact that they are "rare" in the title. Since the transgenic lines may not mark all

cell lines, I think it is appropriate to make this point clear in the manuscript and discussion. To include “rare” in the title more quantitative data or cell line studies would be necessary to provide a robust answer. Instead, I recommend modifying Figure 6 and toning down the statements here.

We have removed 'rare' from the title and reformulated in according to the reviewers' and editor's suggestions.

Figure 6: This figure is overloaded, shows too many details and is redundant. The redundancy is within 6B but also between 6A and 6B. Useful is 6A with the summary of gene expression. 6B is redundant to 6A in terms of cellular differentiation trees, stem cell types (multipotent, unipotent) and experimentally demonstrated and speculative somatic differentiation events (plus: gamete/nerve vs question mark: nematocyte/gland cell/epidermis). One could dispense with 6B altogether if the three variants of multipotent cell populations were only described in the text, which is very clear here. The comments 'poorly-supported' and 'well-supported' for the "primordial germ cell model" and multipotent cell population model" are also part of the discussion and are less helpful for a summary figure, as they are highly dependent on interpretation.

According to the reviewer's suggestion, we have streamlined Fig. 6.

Further comments

Some figure legends are overloaded with text, as they are often linked to statements similar to the main text. Ideally, the statement should be condensed in the figure captions, and the text should only contain descriptions for understanding the experiment.

We have splitted 2 figures (formerly Figs. 3 and 5), which led to the shortening of the respective figure legend. We have also removed the interpretation of flow cytometry results from and shortened descriptions as much as possible without losing information throughout all main figures. The titles of some figure legends have been streamlined and condensed.

In view of the importance of the reticular tract, a brief description of this tissue and its morphogenetic potential should be given at an appropriate place.

We have added information and a sentence to describe the 'reticular tract' in l. 97-100.

Reviewer #2

This manuscript has been considerably clarified and improved in response to the reviewers' comments.

I appreciate the authors' efforts, including removing "stem cell like" from the conclusions and title, and the explicit framing of alternative hypotheses to explain their observations. The new summary diagram (Figure 6) and the overview in a new supplementary table of what observations have (and have not) been done are also helpful.

Specific minor points:

1. There are places where rewording of the added text is needed. As the authors mostly are careful to point out, lack of detection of proteins or mRNA does not necessarily equate to their absence but depends on thresholds of detection. Eg Line 26 and line 141 should be changed eg to "absence of detectable" and "devoid of detectable")
We have added 'detectable' anywhere relevant in the manuscript (l.151, 216).

2. The authors now include in the manuscript quantification of cells with high versus low levels of Piwi and Vasa proteins. This is done by flow cytometry of dissociated cells from [mOr2-Piwi1+] animals. How do the authors know that the [mOr2-Piwi1+] high vs [mOr2-Piwi1+] low cell populations detected this way can be equated to those that they defined by fluorescence microscopy methods? Some justification should be provided, or the assumptions stated explicitly.

We have now added information about our assumption that 'cells with approx. 10-fold higher signal intensity in flow cytometry largely correspond to [mOr2-Piwi1]_{high} identified by confocal microscopy.' (l. 120-124).

3. Word missing on line 374 ('...data from stony corals...')

Corrected.

Reviewer #3

The authors have adequately addressed the points I had raised in the previous review cycle. The manuscript is now very much improved. One minor suggestion for the usage of the term 'germline' in the new title and elsewhere: the term 'germline' is usually used to

refer to segregated germ cell lineages (as seen in ecdysozoans and vertebrates). Here, the authors provide evidence that *Nematostella* possesses adult stem cells that can give rise to gametes and to at least some somatic cells, i.e., its germ cells are not segregated from somatic cells. Therefore, 'gametes' or 'germ cells', but not 'germline', seem more appropriate when discussing non-segregated germ cells. Overall, this manuscript the first major step in characterizing anthozoan stem cells and will have a profound impact on future studies on stem cells and their evolution.

We thank the reviewer for this suggestion, which we have implemented throughout the manuscript.